# Mesospheric gravity wave activity estimated via airglow imagery, multistatic meteor radar, and SABER data taken during the SIMONe–2018 campaign

Fabio Vargas[1], Jorge L. Chau[2], Harikrishnan Charuvil Asokan[2,3], and Michael Gerding[2]

[1]Department of Electrical and Computer Engineering, University of Illinois at Urbana-Champaign, Urbana, IL, USA, 61801
[2]Leibniz Institute of Atmospheric Physics and the University of Rostock, Kühlungsborn, Germany, 18225
[3]Laboratoire de Mécanique des Fluides et d'Acoustique, CNRS, École Centrale de Lyon, Université Claude Bernard Lyon 1, INSA de Lyon, Écully, France

**Correspondence:** Fabio Vargas (fvargas@illinois.edu)

**Abstract.** We describe in this study the analysis of small and large horizontal scale gravity waves from datasets composed of images from multiple mesospheric airglow emissions as well as multistatic specular meteor radar (MSMR) winds collected in early November 2018, during the SIMONe–2018 campaign. These ground-based measurements are supported by temperature and neutral density profiles from TIMED/SABER satellite in orbits near Kühlungsborn, northern Germany (54.1°N, 11.8°E). The scientific goals here include the characterization of gravity waves and their interaction with the mean flow in the mesosphere and lower thermosphere and their relationship to dynamical conditions in the lower and upper atmosphere. We have obtained intrinsic parameters of small and large scale gravity waves and characterized their impact in the mesosphere via momentum flux ($F_M$) and momentum flux divergence ($F_D$) estimations. We have verified that a small percentage of the detected wave events are responsible for most of $F_M$ measured during the campaign from oscillations seen in the airglow brightness and MSMR winds taken over 45 hours during four nights of clear skies observations. From the analysis of small-scale gravity waves ($\lambda_h <$725 km) seen in airglow images, we have found $F_M$ ranging from 0.04–24.74 m$^2$s$^{-2}$ (1.62±2.70 m$^2$s$^{-2}$ on average). However, small-scale waves with $F_M >$3 m$^2$s$^{-2}$ (11% of the events) transport 50% of the total measured $F_M$. Likewise, wave events of $F_M >$10 m$^2$s$^{-2}$ (2% of the events) transport 20% of the total. The examination of large-scale waves ($\lambda_h >$725 km) seen simultaneously in airglow keograms and MSMR winds revealed amplitudes >35%, which translates into $F_M =$21.2–29.6 m$^2$s$^{-2}$. In terms of gravity wave–mean flow interactions, these large $F_M$ waves could cause decelerations of $F_D =$22–41 ms$^{-1}$/day (small-scale waves) and $F_D =$38–43 ms$^{-1}$/day (large-scale waves) if breaking or dissipating within short distances in the mesosphere and lower thermosphere region.

# 1   Introduction

Atmospheric gravity waves represent a class of atmosphere oscillations where buoyancy is the restoring force. These waves transport momentum and energy over large distances within the atmosphere and have as primary sources troposphere disturbances like flow over topography, convective systems, or jets (e.g., Vincent and Alexander, 2020). To preserve kinetic energy, the amplitudes of the gravity waves grow nearly exponentially as they propagate upward into less dense air at higher altitudes. When these waves break and dissipate, they deposit their momentum and energy into the background atmosphere. This affects

the atmosphere over a broad range of scales, from local generation of turbulence to forcing of large-scale circulation (Fritts and Alexander, 2003; Vincent and Alexander, 2020).

This dynamical forcing is most prominent within the mesosphere and lower thermosphere (MLT) at altitudes of typically 50–130 km. Within this range, a large fraction of upward-propagating gravity waves reach their maximum amplitudes and break. The resulting dynamical forcing causes a global-scale circulation within the mesosphere with strong upwelling within

the summer polar region and downwelling within the winter polar region (Houghton, 1978; Holton, 1984). Adiabatic cooling and heating connected to this circulation cause thermal conditions within the mesosphere to deviate far away from radiative equilibrium (Solomon et al., 1987; Vargas et al., 2015).

The role of gravity waves is further complicated as they interact with the background flow as they propagate through the atmosphere. This results in an altitude-dependent filtering of the gravity wave spectrum by the background wind, planetary and

tidal waves. The gravity wave spectrum reaching higher altitudes thus carries an imprint of the dynamics at lower altitudes. Interactions between gravity waves and the mean flow and subsequent wave breaking then generate secondary waves within the mesosphere that propagate both upward and downward. This happens through the creation of temporally and spatially localized momentum and energy fluxes, which successively create strong local body forces and flow imbalances which then excite the secondary waves (Fritts et al., 2006; Vargas et al., 2016; Vadas et al., 2018; Vadas and Becker, 2018; Becker and

Vadas, 2018).

While today the essential nature of the wave-driven circulation of the middle atmosphere is known, important mechanisms and interactions remain to be quantified. Most notably, this concerns wave sources, wave dissipation, and therefore the resulting forcing of the mean flow. A decisive quantity to be specified is the directional $F_M$, including its altitude dependence and its spectral distribution with reference to horizontal ($\lambda_h$) and vertical ($\lambda_z$) wavelengths. Ern et al. (2011) have provided global

distributions of gravity wave $F_M$ in the mesosphere for the first time using global temperature measurements by the Sounding of the Atmosphere using Broadband Emission Radiometry (SABER). They have shown clearly the dependency of gravity wave $F_M$ deposition according with latitude and longitude (non-uniform longitudinal distribution of flux) at different altitude levels from the stratosphere up to the mesosphere along with their seasonal and longer-term variations. Also, attempts of estimating $F_M$ of small-scale, short-period waves using multiple observation platforms such as aircraft, lidar, airglow sounders, radars, and

satellites have been done (e.g, Suzuki et al., 2010; Bossert et al., 2015), while Gong et al. (2019) and Reichert et al. (2019) have relied on lidar, meteor radar, and SABER data to study large-scale, long-period waves perturbing the mesosphere temperature

and the winds simultaneously. These studies report attempts to characterize the wave field and provide $F_M$ estimations of observed events as well.

To bridge gaps in gravity wave dynamics while estimating their $F_M$, an observation campaign named SIMONe–2018 (Spread-spectrum Interferometric Multi-static meteor radar Observing Network) was carried from Nov. 2–9 2018, to collect a large number of specular meteor echoes from several locations (e.g., Vierinen et al., 2019; Charuvil et al., 2020). Also, an all-sky airglow imager system running out of the Leibniz Institute of Atmospheric Physics, Kühlungsborn, Germany, was observing the region in parallel to provide image data of the mesosphere and the horizontal structure of atmospheric oscillations during the campaign.

SIMONe–2018 campaign measurements permit to study distinct spatial and temporal scales of gravity waves perturbing the background wind and the airglow simultaneously. In this paper, we have analyzed all-sky imager (ASI) airglow images and multistatic specular meteor radar (MSMR) wind data to access small-scale as well as large-scale gravity wave dynamics using two different analysis methods for each wave category. Airglow images are processed directly using our auto-detection method for small-scale (<725 km), short-period (<1 hour) gravity waves aided by MSMR background wind measurements for Doppler correction of wave apparent periods ($\tau_o$). For nights presenting obvious large-scale (>725 km), long-period (>1 hour) oscillations, wave features are studied via direct examination of large amplitude wind fluctuations and airglow keogram spectral analysis. We have also obtained measurements of the neutral density, temperature, and OH emission volume emission rates from the Sounding of the Atmosphere using Broadband Emission Radiometry (SABER) instrument (Russell et al., 1999; Mlynczak, 1997) aboard the NASA TIMED (Thermosphere, Ionosphere, Mesosphere Energetics and Dynamics) satellite (http://saber.gats-inc.com) to determine the state of the mesosphere region near the observatory during the campaign. This study shows remarkable instances of waves perturbing the airglow and the wind, providing a singular opportunity to examine the linear gravity wave theory's predictions and the occurrence of gravity waves perturbing multiple mesospheric quantities simultaneously. The main contributions here regard the fraction of observed waves carrying substantial $F_M$ with potential to impart significant changes in the 75-110 km dynamics since we show evidences that most observed waves are likely experiencing dissipation in that region.

## 2 Instrumentation and Data

### 2.1 All-sky airglow imager (ASI)

An all-sky imager(ASI) assembled at Boston University was deployed in late 2016 at the Leibniz Institute of Atmospheric Physics (IAP) in Kühlungsborn, Germany. The imager is equipped with an Andor back-illuminated bare CCD camera and a 30 mm fish-eye lens which record several nightglow emissions over the entire 180° of the night sky. Andor's iKon-M 934 camera is a 1024 x 1024 array and 13 μm pixel pitch with a 13.3 x 13.3 mm active image area. High sensitivity is achieved through a combination of > 90% QE (back-illuminated sensor), low noise readout electronics, and deep TE cooling down to -60°C. The ASI system uses six interference filters enabling the observation of four mesosphere airglow emissions with a background filter for the hydroxyl emission. A filter for the thermospheric redline (at 630.0 nm) is also available, but images of this emission

were not taken during SIMONe–2018 due to filter technical issues. The imaging system operates autonomously via a PC on a nightly basis during moonless periods. Images are obtained on a continuously repeating cycle every ∼2 min with each particular filter accessed every ∼10 min. The specifications of filter wavelengths and integration times are in Table 1. Emission altitudes are discussed in Section 2.3. Preprocessed, low resolution images collected by the ASI are available for visualization at http://sirius.bu.edu/data/. Raw images used in this study are available at https://databank.illinois.edu/datasets/IDB-8585682.

The SIMONe–2018 campaign was carried out for more than a week, but clear skies were seen only during four nights, which limited the optical observations with the all-sky imager. The sky conditions for the four clear nights are summarized in Fig. 1 by zonal and meridional keograms of the O($^1$S) emission. Appendix C discusses in detail how keograms are built from airglow images. The reader is also referred to Vargas et al. (2020) for more keogram analysis information. Although only keograms of the O($^1$S) emission are shown here, we have also built keograms for the other three mesospheric emissions, which are available as supplement files of this paper.

The left-hand side panels of Fig. 1 show keograms built directly from O($^1$S) preprocessed images (Appendix A). The contrast of the images was optimized to show variable features in the brightness present throughout the night. Long-period oscillations seen in the airglow brightness on Nov. 3–4 and Nov. 6–7 keograms indicated by the red ellipses are associated with large-scale, long $\lambda_h$ gravity waves perturbing the greenline layer. For instance, notice in the meridional keogram of Nov. 3–4 the orientation of the brightness variation associated with a large-scale wave in a region tilted from top to bottom during 1930 UTC to 2230 UTC, indicating a coherent oscillation traveling from north to south. The tilt in the brightness region is not pronounced in the zonal keogram for the same time span, indicating a small, negligible wave component in the west-east direction. Perturbations of the same nature are also seen in the O$_2$ and OH emissions for the same nights.

The right-hand side panels of Fig. 1 show zonal and meridional keograms built using time-difference (TD) airglow images. Time-difference operation involves subtracting an image from the previous one (same emission) with the goal of filtering out long-term variations in the airglow brightness (e.g., Swenson and Mende, 1994; Swenson and Espy, 1995; Tang et al., 2005; Vargas, 2019). The result is an image where the contrast of shorter-period, smaller-scale oscillations is enhanced. These small-scale waves show up in the keograms as tilted bright/dark bands. Because long-period waves are suppressed, time-difference keograms permit rapid access to the activity of short-period waves each night.

## 2.2 Multistatic specular meteor radar

During the SIMONe–2018 campaign, MSMR measurements were obtained during seven days continuously. Briefly, the campaign consisted of 14 multistatic links that were obtained by using two pulse transmitters located in Juliusruh (54.63°N, 13.37°E) and Collm (51.31°N, 13.00°E), respectively, and one coded-continuous wave transmitter located in Kühlungsborn. Eight receiving sites were used to receive scattered signal of at least one transmitter. This campaign combines the multistatic approach called MMARIA (Multistatic Multifrequency Agile Radar Investigations of the Atmosphere) (Stober and Chau, 2015) with the SIMONe (Spread Spectrum Interferometric Multistatic meteor radar Observing Network) concept (Chau et al., 2019). In the latter case a combination of spread-spectrum, multiple-input multiple-output, and compressing sensing radar techniques is implemented (Vierinen et al., 2016; Urco et al., 2018, 2019). The winds used in this work have been obtained with a gra-

dient method, i.e., besides the mean horizontal and vertical winds, the gradients of the horizontal components have also been obtained (Chau et al., 2017). Data from one day of this campaign has been used to test a second-order statistics approach by Vierinen et al. (2019). More details of the SIMONe–2018 campaign as well as results of second-order statistics are given in the accompanying paper of this publication (see Charuvil et al., 2020).

Here, we have used the MSMR winds in combination with the airglow data to give a full characterization of the gravity wave dynamics observed during the campaign. Fig. 2 shows the (a) zonal and (b) meridional background winds in the range of 75–105 km measured during SIMONe–2018. Dashed boxes indicate hours of simultaneous operation of the ASI and MSMR systems. The background wind field is calculated from the MSMR measurements using 30-minutes temporal and 1-km spatial windows, respectively. Observe the daily cycle for $z < 80$ km and $z > 100$ km in the plots that is associated with the variation of meteors detections throughout the day; the meteor density is larger at earlier morning hours and smaller at afternoon hours. Because wind calculation relies on the number of meteors to make quality wind estimations, when not enough meteors are detected, the wind cannot be estimated within a reasonable uncertainty level. The background wind is dominated by a 12-hours tidal oscillation presenting amplitudes larger than 50 ms$^{-1}$, but spectral analysis reveals the presence of higher tidal harmonics of 8 and 6 hours (see Fig. 6 and Fig. 7).

Fig. 2c and Fig. 2d also present the zonal and meridional wind fluctuations associated with oscillations caused by gravity waves. To obtain the wind fluctuations, we first average the MSMR raw data over a 400 km$^2$ field of view using a square 4-hour temporal, 4-km vertical window, respectively. Then, we subtract the result from the background wind field. Notice that oscillations of $\tau_o > 4$ hours and $\lambda_z < 4$ km will be suppressed, but not completely eliminated in the resulting wind fluctuations. The fluctuation winds show short-period gravity waves perturbing the wind that are also seen in the airglow. For instance, the oscillation evident in the airglow brightness variation (red ellipse) for Nov. 3–4 (Fig. 1c, meridional keogram) is also evident as coherent oscillations in meridional wind fluctuations (red ellipse) on Nov. 3–4 (Fig. 2d).

## 2.3 Satellite data (TIMED-SABER)

We have also collected observations of the SABER instrument on board the TIMED satellite (Russell et al., 1999; Mlynczak, 1997) within four degrees from the observation site (Fig. 3). The profiles cover the height range from approximately 10 km to more than 100 km. The vertical resolution is $\sim 2$ km. The instrument covers $\sim 52°$ latitude in one hemisphere to $83°$ in the other in a given day. The viewing geometry alternates every 60 days due to $180°$ yaw manoeuvres of the TIMED satellite (Russell et al., 1999). Approximately 1200 temperature profiles are taken each day. SABER publications are available at http://saber.gats-inc.com/publications.php.

SABER profiles used here are presented in Fig. 4a–c, while Fig. 4d shows the calculated volume emission rate of the mesosphere airglow emissions as explained below. The thick lines in Fig. 4 indicate the mean of corresponding individual profiles (dotted lines) for the various orbits of the satellite during the campaign. The corresponding orbits are specified in the legend of each chart.

From Fig. 4a, we can verify that the atmosphere is, in average, stable in the altitude range of 88-99 km since the atmosphere lapse rate ($dT/dz = -3.7$ K/km) is larger than the adiabatic lapse rate ($\Gamma = -9.8$ K/km), and is positive ($dT/dz = 1.6$ K/km)

below and above the 88-99 km range. Even though the satellite orbits registered during SIMONe–2018 were not exactly over the observatory, the instrument measurements are performed in the vertical limb plane that is near or within the field of view of the imager (Fig. 3). Notice that the colored dots indicate where the measurements were made, not the satellite position. Thus, there is a good chance the background atmosphere above the observation site is similar to that indicated by SABER (Fig. 4), although the temperature might still be influenced by gravity waves once we have averaged only a few profiles. Because of that, we are confident using SABER background profiles to make inferences about the propagation conditions for the waves seen over the observatory.

Fig. 4d corresponds to our estimation of volume emission rate (VER) profiles for the OH, $O_2$, and $O(^1S)$ airglow emissions. These VER profiles were calculated using the mean temperature, atomic oxygen, molecular oxygen, and molecular nitrogen profiles in Fig. 4a–b along with the reaction rates of each emission from Vargas et al. (2007). The characteristics of each layer (measured and calculated VERs) are obtained from a Gaussian model (thin lines in Fig. 4d) to fit each profile from which we obtain the layer peak, width, and the full width at half maximum (FWHM). The mean characteristics of the airglow layers are presented in Table 2. The goodness of fitting scores $R^2 > 0.95$ for all five VER curves. The layer centroids, estimated from

$$z_c = \frac{\int z \, \text{VER} \, dz}{\int \text{VER} \, dz},$$

are in general a few kilometers above the estimated layer peaks because of departures of the actual VER vertical structure from the Gaussian fitting model. We have simulated the VER profile for the OH(8,3) using the SABER mean temperature and atomic oxygen profiles for the campaign. The difference between the simulated OH VER and SABER OH VER lies on the averages used as inputs in the VER simulation. However, SABER OH VER and simulated OH VER are much closer in structure if we use individual SABER temperature and oxygen profiles.

## 3  Data Analysis and Results

A full characterization of the gravity wave field requires knowledge of the background wind over the observation site. The significant background wind acting in the vicinity of an airglow layer is a function of the vertical structure of the emission (the VER) that has finite thickness (see Fig. 4d). We take that into account by calculating the weighted background wind (Fig. 6a–c) by using the VER of each layer as weighting functions. The weighted wind expression for a given VER is

$$(u_w, v_w) = \frac{\int (u, v) \, \text{VER} \, dz}{\int \text{VER} \, dz},$$

where $u_w$ and $v_w$ are the weighted zonal and meridional winds (Fig. 6), respectively.

### 3.1  Short-scale gravity wave analysis

We have defined here as small-scale the waves presenting $\lambda_h < 725$ km, while large-scale waves present $\lambda_h > 725$ km. This 725 km threshold corresponds to the length of the diagonal across the field of view of an airglow image mapped into a 512x512 $km^2$ grid. Thus, a 725 km horizontal scale wave would present one crest and one trough fitting the image frame entirely. More details about raw airglow image preprocessing can be found in Appendix A.

The majority of waves observed during SIMONe–2018 are small-scale, fast oscillations of $\tau_i <$1 hour. The keograms of Fig. 1 (right-hand side panels) show the most prominent waves of this category registered during the clear nights of the campaign. These short-scale gravity waves are analysed here using the auto-detection method (Tang et al., 2005; Vargas et al., 2009; Vargas, 2019). The auto-detection method relies on three sequential airglow frames to obtain two time-difference images used in the cross-spectral analysis to obtain gravity wave parameters. The calculation of time-difference images leads to a change in the amplitude of the waves (in the TD images compared to the original ones). The amplitude influences the Fourier analysis and therefore also the result of the cross-correlation. However, this issue is properly taken care of by restoring the amplitude of the waves as seen in the original images. Further details about this correction and the auto-detection method is found in Appendix B.

Fig. 5 shows the results from the auto-detection method for all the emissions recorded during SIMONe–2018. Weighted background winds in Fig. 6a–c were used to carry out the Doppler shift correction on $\tau_o$. Thus, the parameters shown correspond to intrinsic properties of the waves. We have calculated the error bars for the parameters of each wave event measured using the methodology in Vargas et al. (2019). The average error of each parameter is shown in their respective charts in Fig. 5. Since we rely on a set of three images at the time to compute the cross-spectrogram of a set. The time span of each set is about 20 minutes. It is possible that the observed wave events represent waves independent from one another because the observed waves have relatively long vertical wavelength and propagate vertically fast under weak horizontal winds. However, we recognize that this is not always the case and, as the oscillations slow down as they propagate vertically, their residence time within a given airglow layer could be long. Therefore, some of the detected waves could have been counted twice while evaluating the average momentum flux and other wave statistics.

Because every image of a given airglow layer is taken at 10 minutes pace (the filter wheel cycle period), we are only able to resolve wave apparent periods >20 minutes. On the other hand, the exposure time used here is mostly 2 minutes, aliasing could be present due to this relatively long exposure time. However, we have assured the aliasing is minimal in this case because there is no smudging of the small-scale wave structures seen in the images.

The top-center box (Fig. 5c) contains a statistics summary of the measured wave parameters . Fig. 5j shows $\lambda_z$ ranging from 10 to 40 km, while $\lambda_h$ in Fig. 5m clusters around 75–125 km. Waves transporting large $F_M$ are mainly oriented towards Northwest and Southwest (Fig. 5a), but the polar histogram in Fig. 5f shows a large number of waves traveling southeastward into the dominant wind orientation (Fig. 5k). Estimated $\tau_i$ shown in Fig. 5e range within 20–40 minutes, with intrinsic phase speeds in the interval of 30-100 ms$^{-1}$ during the campaign (Fig. 5b). The largest wave relative amplitude estimated from the images is 7% in Fig. 5d, but this does not necessarily translate into large $F_M$ waves, which depends on other wave parameters.

Since the auto-detection method returns wave intrinsic parameters, we are able to estimate $F_M$ of every measured event (e.g., Vargas et al., 2007). Fig. 5l shows the daily $F_M$ of waves detected during SIMONe–2018, with larger $F_M$ waves appearing on Nov. 2–3 and Nov. 3–4. The momentum flux vs. intrinsic wave period chart (Fig. 5g) reveals a tendency of larger $\tau_i$ waves carrying larger $F_M$. Conversely, Fig. 5h (Fig. 5i) reveals that large $\lambda_h$ ($\lambda_z$) waves associate with small $F_M$ quantities.

## 3.2 Large-scale gravity wave analysis

During SIMONe–2018, we have also observed the presence of large-scale gravity waves modulating simultaneously the airglow brightness (Fig. 1c and Fig. 1e) and the horizontal wind (Fig. 2c and Fig. 2d). To study these large-scale oscillations in the wind at the altitude of the airglow, we have calculated the wind fluctuations weighted by the volume emission rate of each layer using

$$(u'_w, v'_w) = \frac{\int (u', v') \, \mathrm{VER} \, dz}{\int \mathrm{VER} \, dz}$$

The result is seen in Fig. 6d–f, where the dashed boxes indicate hours of simultaneous operation of the ASI and MSMR systems.

The weighted wind fluctuations are similar in each layer as the layers peak within $\pm 2$ km from each other (see Table 2) and are thicker (mean FWHM$\sim$15 km) than expected (e.g., Greer et al., 1986; Gobbi et al., 1992; Melo et al., 1996; Wüst et al., 2017). The similarity of these fluctuations is related to larger scale $\lambda_z$ waves seen in Fig. 2c and Fig. 2d. Moreover, because the overlap of the VER profiles is non-negligible, the rms values of the weighted winds fluctuations are expected to have similar magnitude. Calculated rms magnitudes are $6.9\pm1.0$ ms$^{-1}$ and $5.9\pm0.9$ ms$^{-1}$ in the zonal and meridional directions, respectively.

The spectral content of the weighted wind fluctuations is shown in Fig. 7. Several tidal harmonics are still present in the spectra (vertical dotted red lines in Fig. 7). This is due to how the wind fluctuations are calculated, i.e., by first using a 4-hour temporal, 4-km vertical windows to obtain winds representing scales larger, then subtracting the result from the background wind. Thus, the obtained wind fluctuations will contain some of the energy of the tidal modes. However, there are persisting peaks attributed to gravity waves because of their presence in wind fluctuations and keograms. For instance, the peaks in Fig. 7 at the vicinity of 0.24 cycles/hour are seen in the wind fluctuation of Nov. 3–4 (meridional direction). Likewise, Fig. 7 shows a peak near 0.11 cycles/hour corresponding to a wave of $8.9\pm1.0$ hours also seen in the keograms of Nov. 6–7 (zonal direction). A hodograph analysis of the winds must be carried out in a separate work to clarify the nature of the significant peaks in Fig. 7.

We have studied further the wind fluctuations against obvious wave features present in the keograms of Nov. 3–4 and Nov. 6–7. By visual inspection of the images, we have verified that these large-scale waves do not fit within the airglow image field of view (512x512 km$^2$) and are only noticeable via keogram analysis. We carry out the analysis by overlapping the O($^1$S) weighted wind fluctuations on top of the corresponding keograms for these nights (Fig. 8).

On Nov. 3–4 (Fig. 8a), a strong and coherent oscillation is observed in the meridional wind fluctuation while both zonal and meridional keograms present enhanced brightness structures around 2100 UTC (dashed black lines). As the meridional wind fluctuation peaks, the meridional keogram brightness dims (Fig. 8a bottom); as the meridional wind fluctuation reverses direction, the airglow brightens. We have estimated $\tau_o \sim 4.0\pm1.0$ hours for this oscillation from the meridional wind fluctuations, where the assigned uncertainty in $\tau_o$ corresponds to the smallest division in the keogram temporal axis. The tilted brightness structure between 1900 and 2100 UTC in the meridional keogram indicates a wave is traveling southwards. The zonal keogram shows no obvious tilt in the enhanced brightness, suggesting no wave propagation in the east-west direction. That is confirmed from zonal wind (Fig. 8a top) that does not show any apparent oscillation in the same time span.

Similarly, we have observed enhancements in the airglow brightness on Nov. 6–7 associated with a large-scale wave with $\tau_o \sim 8.0 \pm 1.0$ hours estimated from the wave activity in the zonal wind fluctuation seen in Fig. 8b top. The zonal wind fluctuation coincides well with the $O(^1S)$ enhanced brightness structure in the zonal keogram around 0000 UTC. This brightness enhancement shows a slight tilt that indicates a wave propagating from west to east. The negligible brightness tilt in the meridional keogram (Fig. 8b bottom) implies the wave has no evident north-south component.

Spectral analysis of keograms for the two nights showing large-scale waves is in Fig. 9. The zonal and meridional keogram spectra for Nov. 3–4 are in Fig. 9a and Fig. 9c, while zonal and meridional keogram spectra for Nov. 6–7 are in Fig. 9b and Fig. 9d. Appendix C gives further details about the keogram spectral analysis carried out here.

For Nov. 3-4, the zonal keogram spectrum indicates a peak at $k_x = 0$ and $\omega_o = -0.17$ cycles/hour ($\tau_o = 5.7 \pm 1.0$ hours), where the negative sign is associated with a forward evolving time. The meridional keogram spectrum shows a dominant peak at $\omega_o = -0.21$ cycles/hour ($\tau_o = 4.6 \pm 1.0$ hours) and $k_y \sim -0.7 \times 10^{-3}$ cycles/km ($\lambda_y \sim 1365 \pm 136$ km), where the negative sign indicates a southward-propagating wave. The error in $\lambda_y$ is based on Vargas (2019) that estimates 10% error in measurements of large horizontal scale waves ($> 100$ km) from spectral analysis. Notice that because the horizontal scale of the wave in the meridional direction is twice as large as the mapped image FOV ($512 \times 512$ km$^2$), the entire horizontal wave structure is hardly seen in a single airglow image, but is doubtless recognized in the keogram.

The large-scale wave occurring on Nov 6–7 is represented in the zonal spectrum by the peak near $\omega_o \sim -0.11$ cycles/hour ($\tau_o = 9.1 \pm 1.0$ hours) and $k_x \sim 0.2 \times 10^{-3}$ cycles/km ($\lambda_x \sim 4096 \pm 409$ km), where the positive sign indicates an eastward-propagating wave. The meridional keogram spectrum indicates a peak at the same frequency but $k_y \sim 0$, indicating no wave propagation in the meridional direction.

Fig. 10 shows the time-altitude cross-section of the zonal and meridional wind fluctuations for the nights of Nov. 3–4 and Nov. 6–7, respectively. The descending phase progression in time-altitude cross-section reveals these large-scale waves are propagating upwards. We have drawn continuous (dotted) white lines on top of the crests (troughs) of the salient wave structures to estimate $\lambda_z$ and $\tau_o$ of the oscillations. The lines were drawn where the wave structures are better defined on top of the meridional wind (Nov. 3–4) and zonal wind (Nov. 6–7) fluctuation cross-sections. From these lines, we have estimated $\lambda_z = 25.6 \pm 1.0$ km and $\tau_o = 4.3 \pm 1.0$ hours for the wave seen on Nov. 3–4. Notice the assigned error of 1 km in $\lambda_z$ corresponds to $\sim 2$ times the vertical resolution of the vertical axis in Fig. 10, while the assigned error of 1 hour in $\tau_o$ corresponds to the resolution of the temporal axis in Fig. 10.

For the wave seen on Nov. 6–7 we have obtained $\tau_o = 8.0 \pm 1.0$ hours and $\lambda_z = 21.3 \pm 1.0$ km. This long-period wave is not related to the 8-hour tide since the horizontal structure of the oscillation can be seen in the keogram of Fig. 8b entirely. The apparent periods derived here from the descending phase analysis are consistent with those from the keogram spectral analysis shown earlier.

## 4 Discussion

The propagation conditions for gravity waves during SIMONe–2018 are depicted in Fig. 4 showing the temperature and constituent densities near the Kühlungsborn observatory. While the vertical structures of the atomic oxygen density appear normal, the mean temperature indicates convectively favorable conditions for gravity wave vertical propagation as the ambient lapse rate is positive for $z > \sim 87$ km and $z < \sim 99$ km. Within the 88–98 km range, the ambient lapse rate is negative but still sub-adiabatic, and convective instabilities are unlikely to form under these conditions. Thus, gravity wave dissipation due to convective instabilities would not affect the vertical evolution of the gravity wave field during the campaign. We have also verified that dynamic instabilities did not occur because the wind shear was $<30$ ms$^{-1}$/km most of the time during the campaign. On the other hand, because the horizontal winds occasionally achieved relatively large magnitudes $>50$ ms$^{-1}$ (Fig. 2a and Fig. 2b), the wind field could have caused absorption of waves having phase speed $<50$ ms$^{-1}$ traveling to the same direction of the background wind.

We can verify the effect of background wind on the propagation direction of the waves by examining Fig. 5. The momentum flux vs. propagation direction chart (Fig. 5a) shows a number of waves with large $F_M$ oriented towards northwest and southwest, while Fig. 5k shows a dominant southeastward wind during SIMONe–2018 observations. Thus, it is likely that the background wind controls the propagation of southeastward waves via dynamic filtering. However, the wave propagation direction histogram (Fig. 5f) indicates that a significant number of waves still propagate into the wind. These waves must then have horizontal phase speed larger than the background wind. In fact, we have estimated an average $c_i = 56.6 \pm 13.6$ ms$^{-1}$ for waves traveling in the southeast quadrant sector (270° to 360°), while wind has mean magnitude of $39.3 \pm 18.9$ ms$^{-1}$ in the same sector (Fig. 5k. This suggests these fast waves were able to overcome absorption levels while propagating vertically.

Horizontal and vertical wavelengths, intrinsic periods, and intrinsic phase speeds of waves detected during SIMONe–2018 are directly comparable with the results of Li et al. (2011), which used a similar auto-detection method to analyze short-period, fast gravity waves in the airglow. Our statistics show an average $\lambda_z$ of $18.5 \pm 4.6$ km (4.6 km is the sample standard deviation), which is compatible with the results of Li et al. (2011) showing $\lambda_z$ clustering from 20 to 30 km. They have shown $\lambda_h$ clustering around 15–30 km, while our results peak around 75–100 km. Fast waves reported here present remarkably larger horizontal scales than those of Li et al. (2011), which could be associated with the location and type of terrain (Maui–sea vs. Kühlungsborn–land) and gravity wave sources acting near the observatories. Yet, our sample is representative of the winter solstice conditions observed during a week, while that from Maui is representative of the season conditions observed over five years.

There are obvious discrepancies in $\tau_i$ estimated here against those of Li et al. (2011). Observe that $\tau_i$ here bulks around 20 to 30 minutes, while Li et al. (2011) report 77% of waves having $\tau_i < 10$ minutes. We attribute this discrepancy to the different integration time and the filter wheel cycle of the observing airglow camera systems; during SIMONe–2018, we have observed several emissions using a filter wheel cycle period of 10 minutes, which only allows us to detect waves presenting $\tau_o > 20$ minutes. Li et al. (2011) used a filter wheel cycle of two minutes while observing a single emission, allowing detection of waves of time scales as short as $\sim 5$–6 minutes, near the Brunt-Väisälä period. The filter wheel cycle time seems to affect other

parameters as well. For instance, while Li et al. (2011) report a majority of wave intrinsic phase speeds in the range of 50–100 ms$^{-1}$; we have estimated slower intrinsic phase speeds of $31.2\pm17.3$ ms$^{-1}$. The filter wheel cycle influences the sensitivity of the measurement system in the temporal domain, but not the spatial domain, that is, the system will automatically detect faster waves having smaller-periods but in the same $\lambda_h$ range.

In another study, Li (2011) used one year of OH airglow observations over the Andes Lidar Observatory (ALO) in South America to characterize small-scale, fast gravity waves. He found that the peak of distribution of $\lambda_h$ falls in the 20–30 km range, $c_i$ ranges mainly 40–100 ms$^{-1}$ (peak at 70 ms$^{-1}$), 80% of the $\tau_i$ population ranges from 5–20 minutes, and $\lambda_z$ distribution peaks around 15 km. These results resemble those of Maui, and the same discrepancies are applicable for the results in SIMONe–2018. However, the sources of waves over the South American observatory are much clearer and related with convection in central Argentina to the east of ALO. These sources generate fast, short-period, small horizontal scale waves that can be captured over the ALO imager. The farther away the source, the fewer short-periods waves are seen, which explains a secondary peak around $\lambda_h$ =80–100 km shown in Li (2011). This range is comparable to the $\lambda_h$ distribution from the SIMONe–2018 campaign showing a peak around $\lambda_h$ =75–100 km that would be related with tropospheric convective sources active to the north and east of Kühlungsborn during SIMONe–2018.

The momentum flux of high-frequency waves detected during SIMONe–2018 (Fig. 5) is calculated using Vargas et al. (2007, Eq. 13) as showed in Appendix B. The mean momentum flux has a larger component towards the west of -0.36$\pm$1.51 m$^2$s$^{-2}$. Notice the mean $F_M$ shows tendency of a net wave motion westward, while the standard deviation indicates that waves could be moving westward or eastward. The $F_M$ meridional component is $\sim$1/6 of the zonal magnitude. Ignoring for a moment the wave propagation direction, the mean $F_M$ =1.62$\pm$2.70 m$^2$s$^{-2}$. For all the 362 waves detected during SIMONe–2018, 50% of the total $F_M$ is due to waves carrying $F_M$ >3 m$^2$s$^{-2}$ (40 events), that is, only 11% of the detected waves are responsible for 50% of $F_M$ measured during the campaign. This result agrees with the findings of Cao and Liu (2016) that show most of $F_M$ is due to waves that occur very infrequently (low intermittency). However, Cao and Liu (2016) also conclude that small $F_M$ waves are important because of their higher occurrence rate.

Observe that not necessarily the 362 detected events are independent, that is, the same wave could have been detected multiple times throughout the night. However, we rely on sets of three images to make wave detections. Since the time span of each set is about 20 minutes (10 minutes between successive images), waves detected in the given set would have disappeared from the imager FOV after that time span. This is because the duration of quasi-monochromatic wave packets in airglow images are generally short. Thus, the same waves are unlike be detected in multiple image sets. This way, it would be more likely that most of the 362 detections correspond to different, independent waves.

In spite of the small mean value, $F_M$ bursts between 10–30 m$^2$s$^{-2}$ were mainly seen in the O$_2$ emission during the campaign. These waves were traveling northwestward with $\tau_i$ =30–40 minutes, $\lambda_h$ $\sim$90 km, and $\lambda_z$ =12–15 km (see charts of each emission in the supplement files of this publication). The sum of $F_M$ of these waves (8 events) accounts for 20% of the total small-scale wave $F_M$ measured during the campaign. It is not clear why the enhanced waves are seen most in the O$_2$ emission once the layer's peaks nearly overlap, but could be related to that the O$_2$ VER having a narrower FWHM (see Table 2). This

way, shorter $\lambda_z$ waves would be detected primarily in $O_2$ images, presenting larger $F_M$ since it increases as $\lambda_z$ decreases (Fig. 5i).

We see that even in smaller numbers, the more energetic, larger $F_M$ waves could have greater impact in the atmosphere. For instance, Bossert et al. (2015) investigated, during the Deep Propagating Gravity Wave Experiment (DEEPWAVE), mountain waves presenting horizontal scales of 200–300 km with $F_M$ in the range of 20-105 $m^2s^{-2}$. Similarly, Smith et al. (2020) estimated $F_M \sim 232$ $m^2s^{-2}$ associated with an extensive and bright mesospheric gravity wave event seen over the El Leoncito Observatory, Argentina (31.8° S, 69.3° W), during the nights of 17 and 18 March 2016. The waves observed in this study

carrying $F_M > 3$ $m^2s^{-2}$ would potentially cause $F_D \sim$ 22–41 $ms^{-1}$/day (Vargas et al., 2007, Fig. 9e), considering that the wave breaking continues for 24 hours. This would lead to considerable mean flow deceleration and body forces capable of exciting secondary waves as point-like sources (Vadas and Becker, 2018). Considering the wave source and wave breaking mechanism acting for 4 hours (about half of a typical nighttime observation period) in a given altitude, we estimate a potential mean flow deceleration of 3.7–6.8 $ms^{-1}$ in this time span (4 hours) due to wave forcing.

In a similar study, Suzuki et al. (2010) presented identical gravity wave structure detected in airglow intensity, radar wind, and lidar temperature. In airglow keograms from northern hemisphere stations in Japan, they observed small-scale gravity waves with $\lambda_h \sim 170$ km, period of 1 hour propagating northeastward at $\sim 50$ $ms^{-1}$. Using from both airglow images and meteor radar wind, they have calculated an average $F_M$ of 0.8 $m^2s^{-2}$ at 94 km and 1.5 $m^2s^{-2}$ at 86 km for the observed oscillations. The Suzuki et al. (2010) flux measurements agree with our estimates for small-scale waves that show a majority of events carrying small $F_M$. They have also estimated the acceleration of 0.8 $ms^{-1}$/hour (19.2 $ms^{-1}$/day) at the 94 km height,

which is close to our estimations of $F_D$ for small-scale waves.

Ern et al. (2011) shows absolute $F_M$ values of $\sim 10^{-3.9}$ Pa at 50 km altitude and $\sim 10^{-4.3}$ Pa at 70 km altitude in the northern hemisphere in January for latitudes/longitudes near the Kühlungsborn observatory, evidencing momentum flux deposition in the middle atmosphere. Thus, it is likely that small-scale waves observed here are mostly dissipating as they travel through the

MLT, in agreement with Vargas et al. (2019). In other flux estimation study using airglow imagery of gravity waves, Vargas et al. (2009) has revealed $F_M$ ranging from $\sim 1.5$ to $\sim 4.5$ $m^2/s^2$, while radar measurements of (e.g, Yuan and Fritts, 1989) estimated $F_M = 5$–15 $m^2/s^2$. Also, it is believed 70% of the momentum is carried by short-period waves (<1 hour) (Vincent, 1984). Estimations of $F_D$ (wave drag) in the meridional direction from airglow measurements unveiled accelerations of 3 $ms^{-1}$/day (Vargas et al., 2015), which is significant given that the meridional wind magnitude is weak ($\sim 20$ $ms^{-1}$ or less at

mid latitudes), while in the zonal wind the wave $F_D = 15$–60 $ms^{-1}$/day (Vincent and Fritts, 1987).

We have also estimated the horizontal wavenumber and apparent frequency of the large-scale waves shown in the airglow (Fig. 8) from the spectrum of the zonal and meridional keograms in Fig. 9. We then estimate $\lambda_z$ of the events assuming a Brunt-Väisälä period of 5.5 minutes ($N = 0.01904$ rad/sec) and an inertial period of 14.8 hours ($f = 0.11816 \times 10^{-3}$ rad/sec) for the Kühlungsborn latitude. We use the acceleration due to gravity $g = 9.5$ m/s$^2$ for the mesosphere.

The wave occurring on Nov. 3–4 presents $k_h = k_y = -0.7 \times 10^{-3}$ cycles/km and $\omega_o = 0.215$ cycles/hour estimated from the keogram spectra. Weighted background wind field over the observatory at 2315 UTC presented $u = 28.5$ $ms^{-1}$ and $v = -1.4$ $ms^{-1}$ at the instant the wave was in the dimmer phase of its cycle in the airglow. Applying then Doppler shift correction, we

estimate an intrinsic frequency $\omega = 0.211$ cycles/hour for the wave. Finally, using the dispersion relation (Appendix C), we derive $\lambda_z = 25.1 \pm 2.5$ km for the Nov. 3–4 wave, where the uncertainty is 20% as estimated in Vargas (2019). This wavelength compares well with $\lambda_z = 25.6 \pm 1.0$ km obtained by visual inspection of Fig. 10.

Likewise, the Nov. 6–7 wave has $k_h = k_x \sim 0.2441 \times 10^{-3}$ cycles/km and $\omega_o = 0.11$ cycles/hour. Applying once again the Doppler shift correction using background wind components of $u = 26.0$ ms$^{-1}$ and $v = -30.0$ ms$^{-1}$ at 2326 UTC, we obtain an intrinsic frequency of $\omega = 0.087$ cycles/hour. From the dispersion relation we then estimate $\lambda_z = 20.5 \pm 2.0$ km for this wave, which also agrees with the measured value of $\lambda_z = 21.3 \pm 1.0$ km from Fig. 10.

The amplitude of the large-scale from keogram waves are $I'_\% = 36.5\%$ (Nov. 3–4 @ 2130 UTC) and $I'_\% = 47.9\%$ (Nov. 6–7 @ 0030 UTC). These amplitudes are relative to the mean airglow brightness of each night. As demonstrated in Appendix B, the vertical wavelength of each wave along with their perturbations in brightness permit to evaluate their relative perturbation in temperature as $T'_\% = 9.1\%$ (Nov. 3–4) and $T'_\% = 13.7\%$ (Nov. 3–4). Then, we finally estimate $F_M$ (see Appendix B) by using the intrinsic parameters found for the observed large-scale waves, which are $F_M = 21.2$ m$^2$s$^{-2}$ for the wave seen on Nov. 3–4, and $F_M = 29.6$ m$^2$s$^{-2}$ for that seen on Nov. 6–7. Table 3 shows a summary of the main features of the large-scale waves as discussed above. We expect the uncertainties in $F_M$ to be large (>40%) given that the $F_M$ variables incur in uncertainties that are transferred to $F_M$ via error propagation (Vargas, 2019).

Based upon $F_M$ values of the large-scale waves, we estimate for the southward-traveling wave (Nov. 3–4) a momentum flux divergence $F_D \sim 43$ ms$^{-1}$/day in the meridional flow, assuming this wave breaks or dissipates in a given level along its vertical path. Similarly, the Nov. 6–7 wave would cause a deceleration of $F_D \sim 38$ ms$^{-1}$/day in the zonal flow in the breaking or dissipation level. These large-scale, large amplitudes waves would have a greater impact on the mean flow than small-scale waves, even though these waves are less frequent in mesospheric measurements than their small-scale counterparts.

Gong et al. (2019) have also investigate the properties of large-scale, long-period waves observed on May 30, 2012 in China. Datasets of three instruments used in the study have shown evidences of a same gravity wave perturbing lidar and SABER temperatures as well as meteor radar winds. The parameters associated with the observed wave are $\lambda_h = 560$ km, $\lambda_z = 8$–10 km, $\tau_o = 6.6$–7.4 hours, and phase speed of 21 ms$^{-1}$. Gong et al. (2019) and Reichert et al. (2019) along with our study represent few efforts to characterize larger-scale gravity waves propagating from the stratosphere into the mesosphere using multi-instrument datasets.

According to Vargas et al. (2019), only a minority of waves seen in the airglow($\sim 5\%$) are in non-dissipating regime. Vargas et al. (2019) also shows that the majority of the gravity waves present strong dissipation and transfer momentum flux to the main flow within a distance of two atmosphere scale heights (12-14 km). Thus, large $F_M$ waves discussed here are likely to present dissipative or breaking characteristics given their larger amplitudes. This is not without controversy, since recent radar measurements in Antartica (Sato et al., 2017) have shown longer-period gravity waves (1 hour–1 day) transporting larger $F_M$, although short-period oscillations also have significant $F_M$ but relatively smaller.

Recently, Vadas and Becker (2018) have modeled the evolution of mountain waves over the Antarctic Peninsula after observational results of large-scale, long-period waves seen in the mesosphere (Chen et al., 2013, 2016) attributed to an unbalanced flow in the lower stratosphere. This imbalance excited upward (downward) propagating oscillations from the knee of fishbone-

like structures at 40 km altitude, which are associated with the excitation of secondary waves from the breaking of extensive mountain wave structures. Although other modeling efforts also attribute the excitation of non-primary waves to localized turbulence eddies from gravity wave breaking (e.g., Heale et al., 2020), we believe that the large-scale waves observed in this study are the product of the Vadas and Becker (2018) mechanism at play in the stratosphere. In fact, preliminary analysis of temperature profiles at 0–90 km altitude acquired by the IAP Rayleigh Lidar system on Nov. 6–7 revealed fishbone structures at 40-45 km, resembling the predictions of Vadas and Becker (2018). We will investigate in detail the possible connection with the large-scale waves seen here in a separate paper; specifically, we want to identify potential sources of primary waves in the vicinity of Kühlungsborn during SIMONe–2018, and also trace the observed large-scale waves back to their excitation altitude around the fishbone knee region at 40-45 km revealed in the filtered lidar temperatures.

## 5    Conclusions

In this paper, gravity waves of small and large horizontal scales were characterized by their intrinsic wave parameters, amplitudes, momentum fluxes, and momentum flux divergences. We have focused the analysis on data recorded simultaneously by an airglow all-sky camera, multistatic specular meteor radar, and TIMED/SABER satellite to obtain a more extensive collection of complementary information about the state of the mesosphere region over the observatory during the campaign. To uncover small horizontal scale features, we have used an auto-detection method to process all-sky airglow images and background meteor radar winds. Large-scale waves were characterized by spectral analysis of airglow keograms and altitude vs. time cross-section of wind fluctuations.

Our results indicate that 11% of all detected gravity wave events have large amplitudes and carry 50% of the total $F_M$ estimated during SIMONe–2018. These fewer wave events could impart mean flow deceleration of $F_D =21$–$43$ ms$^{-1}$/day towards the wave propagation direction at breaking or dissipation levels. We have estimated $F_D$ using Vargas et al. (2007, Fig. 9e) results for waves having $\lambda_z = 20$–$25$ km and $\lambda_h >100$ km. However, the deceleration will be smaller because the waves are unlikely to be breaking or dissipating continuously for 24 hours, dying out earlier.

Given the relatively large $\lambda_z$ and $\lambda_h$ of the observed large-scale waves, there is a possibility that these events are the product of secondary wave excitation via the mechanism identified by Vadas and Becker (2018). This possibility is supported by stratosphere fishbone structures uncovered in filtered temperatures collected over Kühlungsborn with the IAP Rayleigh Lidar system on Nov. 6–7. A complete analysis of these structures will be given in a separate paper, in which we also plan to show the origin of the primary waves in the troposphere from weather images as well as the presence of non-primary waves in other datasets such as that of the AIRS on board the AQUA satellite.

## Appendix A:  Airglow Image Preprocessing

For a given observation night, in the absence of contamination sources (e.g., cloudiness), a series of airglow images is produced by our airglow imager system with 10 minutes per image and ∼2 minutes integration time. Prior to carry out spectral analysis,

each raw airglow image must pass through a series of preprocessing steps. First, the image frame is centralized such as the image zenith coincides with the central pixel of the image frame. Second, the image is rotated and flipped over such as the image top points northward and the image left points eastward. Third, the stars are removed using a star suppression algorithm Tang et al. (2005). Forth, the resulting image is then mapped onto a geographic plane of 512x512 km$^2$ projected at the height of the emission layer Garcia et al. (1997). Fifth, the images are detrended by subtracting a fitted linear surface from the image frame. After these preprocessing steps, the resulting frames are then uniform across the FOV with a pixel resolution of 1 km/pixel and ready for spectral analysis (auto-detection or keogram spectral analysis methods) to obtain gravity wave parameters present in the images.

## Appendix B: Auto-Detection Method

The auto-detection method for image processing and analysis was used in this study to obtain parameters of quasi-monochromatic waves from sequences of airglow images. This process detects waves and estimates its parameters automatically, making the study of gravity waves more effective, especially in relation to the estimation of $F_M$. Compared to conventional techniques, which involve looking for waves from visual inspection of preprocessed image sequences, this method is more optimized because it processes a set of three images at a time, requiring relatively less processing time.

The $F_M$ carried by vertically propagating waves are estimated from intrinsic parameters waves, knowing the prevailing wind calculated from meteoric radar data. Preprocessed images mapped in a 512x512 km$^2$ grid are cropped around the zenith to produce the 174x174 km$^2$ analysis window because the central region of the image is less sensitive to lens distortion.

The method corrects automatically $\tau_o$ Doppler shift due to the background wind. This is done by shifting the image pixels of each direction by a distance proportional to the wind velocity divided by the image acquisition period (Tang et al., 2005). Pixel shifting is performed in the first and last images of a set. The corrected set are used to compose two TD images. A TD image is produced by subtracting an image from the previous one in the image set (Swenson and Mende, 1994; Swenson and Espy, 1995; Tang et al., 2005; Vargas, 2019).

Two TD images are generated from sets of three consecutive preprocessed airglow images around a given instant. The Fourier transform is applied to each TD image, and the cross-spectrum is then obtained from the individual TD spectra. Thus, the spectrograms of each TD image are obtained from the 2D-FFT transform, which, in turn, are combined to form the cross-spectrogram of the image set (e.g., Vargas, 2019).

Let $J_1(k_x, k_y)$ and $J_2(k_x, k_y)$ be the Fourier transforms of two TD images from a given set. In general, lateral lobes associated with spectral peaks appear in the spectrogram as a result of the limited spatial extent of the image. In this work, we applied the 2D Hanning window in the TD images to minimize the lateral lobes while preserving the energy of sinusoidal components associated with gravity waves. The cross-spectrogram is described in terms of both $J_1(k_x, k_y)$ and $J_2(k_x, k_y)$ as

$$I_{1,2} = \frac{J_1(k_x, k_y) J_2^*(k_x, k_y)}{n^2}$$

where the asterisk designates the complex conjugate and $n^2$ the number of pixels in the image. The cross-spectrogram contains information about the wavenumber, temporal frequencies, and the phase difference of the dominant components of the spectrum.

The dominant wavenumbers of the image set are then identified from the amplitude cross-spectrogram $|I_{1,2}|$, while the dominant wave periods are determined from the phase cross-spectrogram. The wavenumbers $k_x$ and $k_y$ are determined at the location of the $i^{th}$ spectral peak to obtain $\mathbf{k_h} = (k_x, k_y)$, providing $\lambda_h = 1/(k_x^2 + k_y^2)^{\frac{1}{2}}$. The wave orientation is then $\phi = \tan^{-1}(\frac{k_y}{k_x})$.

From the phase cross-spectrogram we obtain the phase shift $\delta\theta$ of the wave between TD images at the location of the spectral peak $(k_x, k_y)$. We now can estimate the $c_i$ of the wave using

$$c_i = \frac{1}{2\pi} \frac{\delta\theta}{\delta t} \lambda_h,$$

where $\delta t = 10$ minutes the filter wheel cycle period. The intrinsic wave period is then found from $\tau_i = \frac{2\pi}{\omega} = \frac{\lambda_h}{c_i}$. Notice that at this point, the wave propagation direction $\phi$ has an 180° ambiguity. This ambiguity is resolved by taking the $(k_x, k_y)$ pair values from the phase cross-spectrogram where $\delta\theta < 0$, which corresponds to a time coordinate progressing forward.

The airglow brightness $I$ detected by the CCD sensor can be considered as the superposition of a basic state $\bar{I}$ and a state disturbed by waves $I'$ such as $I = \bar{I} + I'$. Again, $I'$ is estimated in the wavenumber domain based on the amplitude cross-spectrogram. The undisturbed component $\bar{I}$, on the other hand, is obtained by the mean airglow brightness over the image field of view.

The relative wave amplitude in intensity $I'_\% = 100 \times (\frac{I'}{\bar{I}})$ over the FOV is estimated by integrating the energy around a given spectral peak. The wave information is only stored if $I'_\%$ has energy >10% of the total cross-spectrogram energy. The basic hypothesis for restoring the wave energy is that the wave content throughout the image is uniform. However, the animation of TD images reveals that monochromatic waves do not always cover the entire FOV, thus, the energy extracted from the wavenumber domain represents an average wave energy over the FOV. The size of the analysis window (174x174 km$^2$) is important because it is small enough to allow the wave event cover the entire FOV, giving a more accurate estimate of its energy, and large enough to ensure the detection of waves in the range of 2–174 km. Notice that while this procedure restricts the field of view, dynamic parameters of gravity waves can be estimated more reliably since the full wave structure is captured by this smaller analysis window.

The TD image operation affects the amplitude of the waves of different periods according to the following equation

$$I'_{TD}/I' = 2\sin(\frac{\omega\delta t}{2})$$

where $I'_{TD}$ is the wave amplitude after the TD operation is carried out and $\delta t$ the image acquisition time. Fig. 11 shows this effect were short period waves will be amplified ($I'_{TD} > I'$) while long period waves will be attenuated ($I'_{TD} < I'$). The amplification range lies within $1.2\delta t < 2\pi/\omega < 6\delta t$ minutes. Prior to carry out other e have to obtain the real wave amplitude prior to calculate other parameters. Fig. 11 points out a filtering effect intrinsic to the auto-detection method that limits the observation of waves having periods that are harmonics of the acquisition time (that is, 10 min, 5 min.). Observe in Fig. 11

that the amplitude decays fast for waves with a period <20 min and becomes zero for a period equal to 10 min. Ideally, a better airglow experiment would rely on having individual all-sky imagers dedicated to individual emissions. Having a higher time resolution between sequential images is ideal, but this can not be accomplished in most of the airglow observatories. This difficulty manifests here as the auto-detection method's inability to capture waves of periods shorter than the integration time, making our results restricted to a subset of observable gravity wave periods.

We then evaluate $\lambda_z = \frac{2\pi}{m}$ using the complete gravity wave dispersion relation

$$m^2 = \frac{(N^2 - \omega^2)}{(\omega^2 - f^2)} k_h^2 + \frac{\omega^2}{\gamma g H} - \frac{1}{4H^2}$$

where $\gamma = c_p/c_v$ the ratio of specific heats and $H$ the scale height in the MLT, while $g$, and $N$ are the acceleration due to gravity and the Brunt-Väisälä frequency, respectively.

The momentum flux is calculated using Vargas et al. (2007, Eq. 13)

$$F_M = -\frac{1}{2} \frac{\omega^2 g^2 m}{N^4 k_h} |\frac{T'_\%}{100}|^2,$$

where $\omega$, $k_h$, $m$, $T'_\%$, $g$, and $N$ are the wave angular intrinsic frequency, horizontal wavenumber, vertical wavenumber, percent temperature fluctuation, acceleration due to gravity, and the Brunt-Väisälä frequency, respectively. The percent temperature fluctuation $T'_\%$ is calculated using the cancellation factor Vargas et al. (2007, Eq. 12)

$$CF = \frac{I'_\%}{T'_\%} = 4.6 - 3.7 e^{0.006(\lambda_z - 6)},$$

as the relative wave amplitude in intensity $I'_\%$ is obtained from the amplitude spectrogram as described earlier.

The operations above run in a loop that iterates continuously for the the number of images collected in a given observation night. The wave parameters, their uncertainties, and the occurrence time stamps of the events are stored in a separated file for each night.

## Appendix C: Keogram Spectral Analysis

While individual airglow images represent a routine way to study short-period, small-scale gravity waves, keograms are conveniently used in this study to investigate the characteristics of major low-frequency, large-scale oscillations, revealing wave activity over the time span of the observation night.

We built zonal (meridional) keogram by taking the central row (column) of raw or preprocessed airglow images collected in a given observation night (Vargas et al., 2020). In the zonal keogram, the vertical scale indicates west (negative) and east (positive), while the vertical scale in meridional keograms indicates south (negative) and north (positive). The horizontal scale in both zonal and meridional keograms refers to the universal coordinated time (UT). Notice the center of vertical axis of the keograms corresponds to the brightness registered by the zenith pixel localized at the center of the images.

Large and small scale waves show up in keograms as tilted luminous or dark patches. The deeper the tilt is, the slower the phase speed (long $\tau_o$) of the wave (Vargas et al., 2020). The horizontal wavelength can be also determined from keogram

images as long the wave has nonzero phase speed at least in a given direction (zonal or meridional). The wave tilt angle is measured from the horizontal axis to the wave luminous patch in the keogram, and is positive if the wave travels eastward (northward) in the zonal (meridional) keogram.

Zonal and meridional keograms are airglow brightness time series as a function of zonal and meridional distances. The temporal axis has resolution of 10 minutes and the spatial axis (zonal and meridional) have resolution of 1 km/pixel. Thus,

waves presenting $\tau_o > 20$ minutes and $\lambda_h > 2$ km can be resolved by this method. The spectral analysis of keograms is carried out in the Fourier space via 2D-FFT preceded by Hanning windowing. The spectral content of the zonal (meridional) keogram can be seen in Fig. 9a (Fig. 9c). To obtain the wave parameters from the keogram spectrum, the wavenumbers of higher energy are selected in the range of $\omega_o < 0$ only, which corresponds to time progressing forward. Notice that by considering $\omega_o < 0$, the ambiguity in the wave propagation direction is resolved.

The i$^{th}$ spectral peak in the spectrogram are pairs $(k_x, \omega_o)$ and $(k_y, \omega_o)$ from the zonal and meridional keograms spectrum, respectively. These pairs correspond to parameters of prominent large-scale waves seen in the keograms. Here, $k_x$, $k_y$ are the zonal and meridional wavenumber components of $\mathbf{k_h} = (k_x, k_y)$ from where we obtain $\lambda_h = 1/(k_x^2 + k_y^2)^{\frac{1}{2}}$. The apparent wave frequency is $\omega_o$, from where we obtain the apparent wave period $\tau_o = 1/\omega_o$. Notice that $\tau_o$ can be determined from both zonal or meridional keogram spectrum since the temporal axis is common to both of them.

The intrinsic frequency $\omega$ is determined using background winds from meteor radar projected in the direction of wave propagation. This dependency is described by $\omega = \frac{2\pi}{\tau_o} - \mathbf{k_h} \cdot \mathbf{v}$, where $\mathbf{v} = (u, v)$, $u$ and $v$ are the background wind components in the zonal and meridional directions, respectively.

We can combine the observed frequency with the observed background wind to derive $\tau_i$ of the waves using $\omega = \omega_o - \mathbf{k_h} \cdot \mathbf{v}$, where $\omega_o$ is the apparent frequency measured by an observer on the ground, $\mathbf{k_h} = (k_x, k_y)$ and $\mathbf{v} = (u, v)$ are the horizontal

wavenumber and wind vectors with components oriented in the zonal and meridional directions, respectively.

We then estimate $\lambda_z$ of the events by applying the simplified gravity wave dispersion relation

$$m^2 = \frac{(N^2 - \omega^2)}{(\omega^2 - f^2)} k_h^2,$$

where $m = 2\pi/\lambda_z$ is the vertical wavenumber, $N$ is the Brunt-Väisälä frequency, and $f$ the inertial frequency. We have omitted the term $1/4H^2$ in $m^2$ equation as it causes only 5% difference on the derived $\lambda_z$.

Finally, we derive $F_M$ for large-scale waves seen in keograms by evaluating Eq. 12 and Eq. 13 of Vargas et al. (2007) in a similar fashion as shown in Appendix B, although the keogram spectral analysis is noniterative.

*Author contributions.* FV devised the data processing methods, carried out data analysis. JLC conceived SIMONe and ran the campaign. HCA provided preprocessed meteor radar wind data. MG provided lidar data and revised the manuscript. All authors contribute writing different parts of the manuscript.

*Competing interests.* None

*Acknowledgements.* This work was partially supported by the Deutsche Forschungsgemeinschaft (DFG, German Research Foundation) un-
der SPP 1788 (DynamicEarth)-CH1482/2-1 and MSGWaves/PACOG project LU 1174/8-1. Fabio Vargas' research received external funding
from the National Science Foundation under NSF AGS grant 17-59573 and NSF AGS grant 19-03336. Multistatic specular meteor radar
winds used in this study can be downloaded from https://databank.illinois.edu/datasets/IDB-8585682. We also want to acknowledge here the
SABER team for making available the data used in this publication at http://saber.gats-inc.com/browse_data.php#. We thank the support of
Boston University colleagues and in particular J. Baumgardner to operate the airglow imager.

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

**Table 1.** Configuration of the All-sky imager system used to collect airglow images during the SIMONe–2018 Campaign. Airglow images of the campaign are available at http://sirius.bu.edu/data/.

| Filter | Emission | Wavelength (nm) | Integration Time (sec) |
|--------|----------|-----------------|------------------------|
| RG695 | OH | 695.0–1050.0 | 15 |
| 6050C | Background OH | 605.0 | 120 |
| 5893C | NaD | 589.3 | 120 |
| 8660C | $O_2(0,1)$ | 864.5 | 120 |
| 5577C | $O(^1S)$ | 557.7 | 120 |
| 6300C | $O(^1D)$ | 630.0 | 120 |

**Table 2.** Centroid, peak, and FWHM of the OH, $O_2$, and $O(^1S)$ layers measured and calculated using TIMED/SABER data collected near Kühlungsborn during SIMONe–2018 campaign.

| Emission | Wavelength | Origin | Layer Centroid (km) | Layer Peak (km) | FWHM (km) |
|---|---|---|---|---|---|
| OH(A) | 2.1 $\mu$m | SABER | $\sim$87.3 | $\sim$86.4 | $\sim$14.3 |
| OH(B) | 1.6 $\mu$m | SABER | $\sim$85.8 | $\sim$84.8 | $\sim$12.5 |
| OH(8,3) | 727.3 nm | simulation | $\sim$89.4 | $\sim$86.5 | $\sim$18.7 |
| $O_2$(0-1) | 864.5 nm | simulation | $\sim$91.1 | $\sim$88.0 | $\sim$14.6 |
| $O(^1S)$ | 557.7 nm | simulation | $\sim$93.3 | $\sim$91.4 | $\sim$16.7 |

**Table 3.** Estimated features of the large-scale waves observed in the airglow and meteor radar wind data.

| Date | $\lambda_h$ (km) | $\lambda_z$ (km) | $\tau_o$ (h) | $\tau_i$ (h) | $c_o$ (ms$^{-1}$) | $c_i$ (ms$^{-1}$) | $I'$ (%) | $T'$ (%) | $F_M$ (m$^2$s$^{-2}$) |
|---|---|---|---|---|---|---|---|---|---|
| Nov. 3–4 | 1365±136 (Fig.09) | 25.1±2.5 (Fig.09) 25.6±1.0 (Fig.10) | 4.0±1.0 (Fig.08) 4.6±1.0 (Fig.09) 4.3±1.0 (Fig.10) | 4.7±1.0 | 81.6±19.5 | 80.3±19.5 | 36.5±3.6 | 9.1±1.8 | 21.2±8.4 |
| Nov. 6–7 | 4096±409 (Fig.09) | 20.5±2.0 (Fig.9) 21.3±1.0 (Fig.10) | 8.0±1.0 (Fig.08) 9.1±1.0 (Fig.09) 8.0±1.0 (Fig.10) | 11.5±1.0 | 125.2±18.6 | 99.1±18.6 | 47.9±4.8 | 13.7±2.7 | 29.6±11.8 |

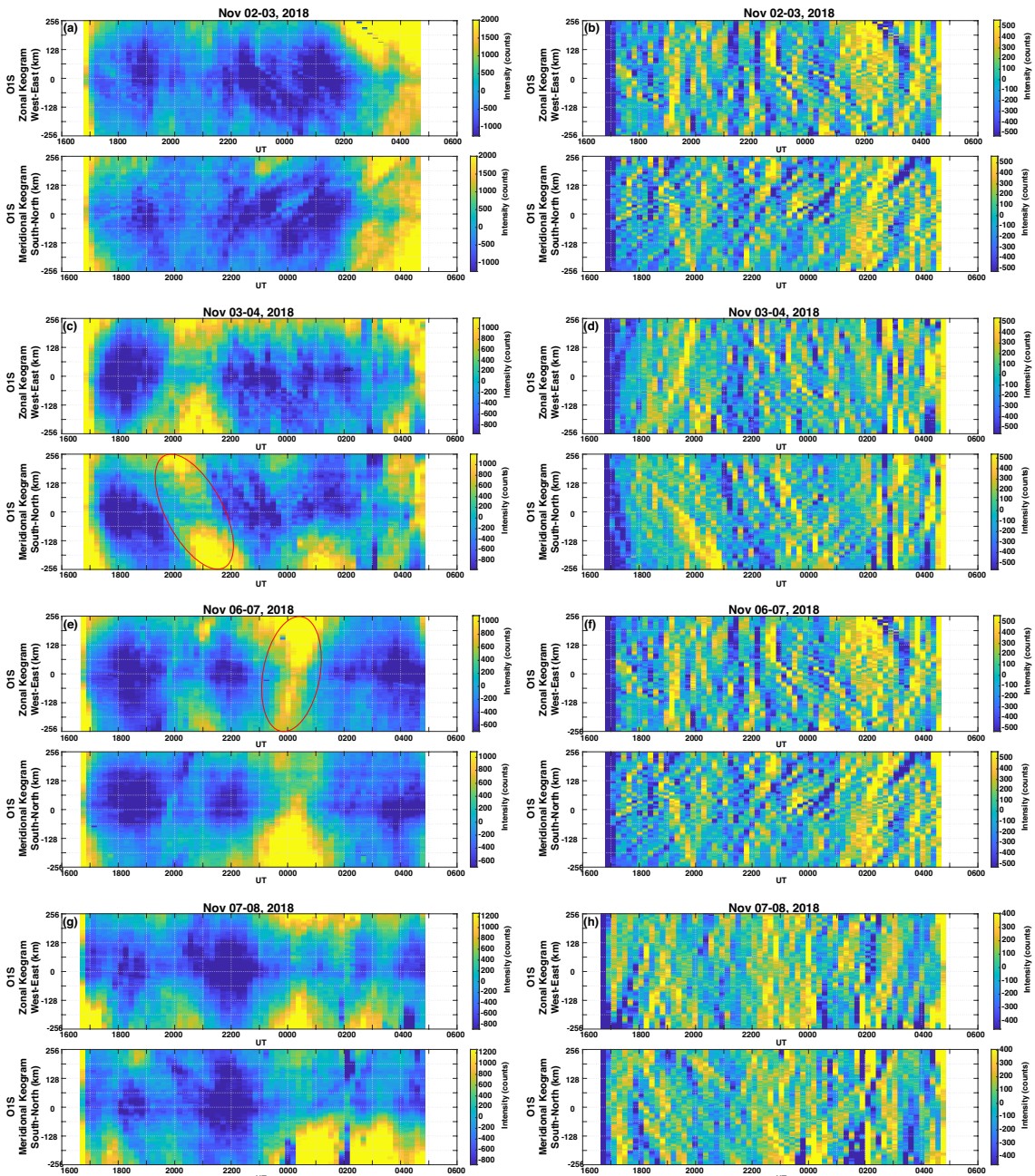

**Figure 1.** Composite keograms of O($^1$S) airglow images taken on clear nights during the SIMONe–2018 campaign. The keograms in panels (a), (c), (e), and (f) were built using light frame images, while keograms in panels (b), (d), (f), and (g) were built using TD images. Time-difference keograms show short-period waves in higher contrast, while light frame keograms show mainly long-period oscillations. Note the enhanced airglow brightness (red ellipses) on Nov. 3–4 (meridional keogram) and Nov. 6–7 (zonal keogram) associated with large-scale gravity waves also seen in wind fluctuations of Figure 2.

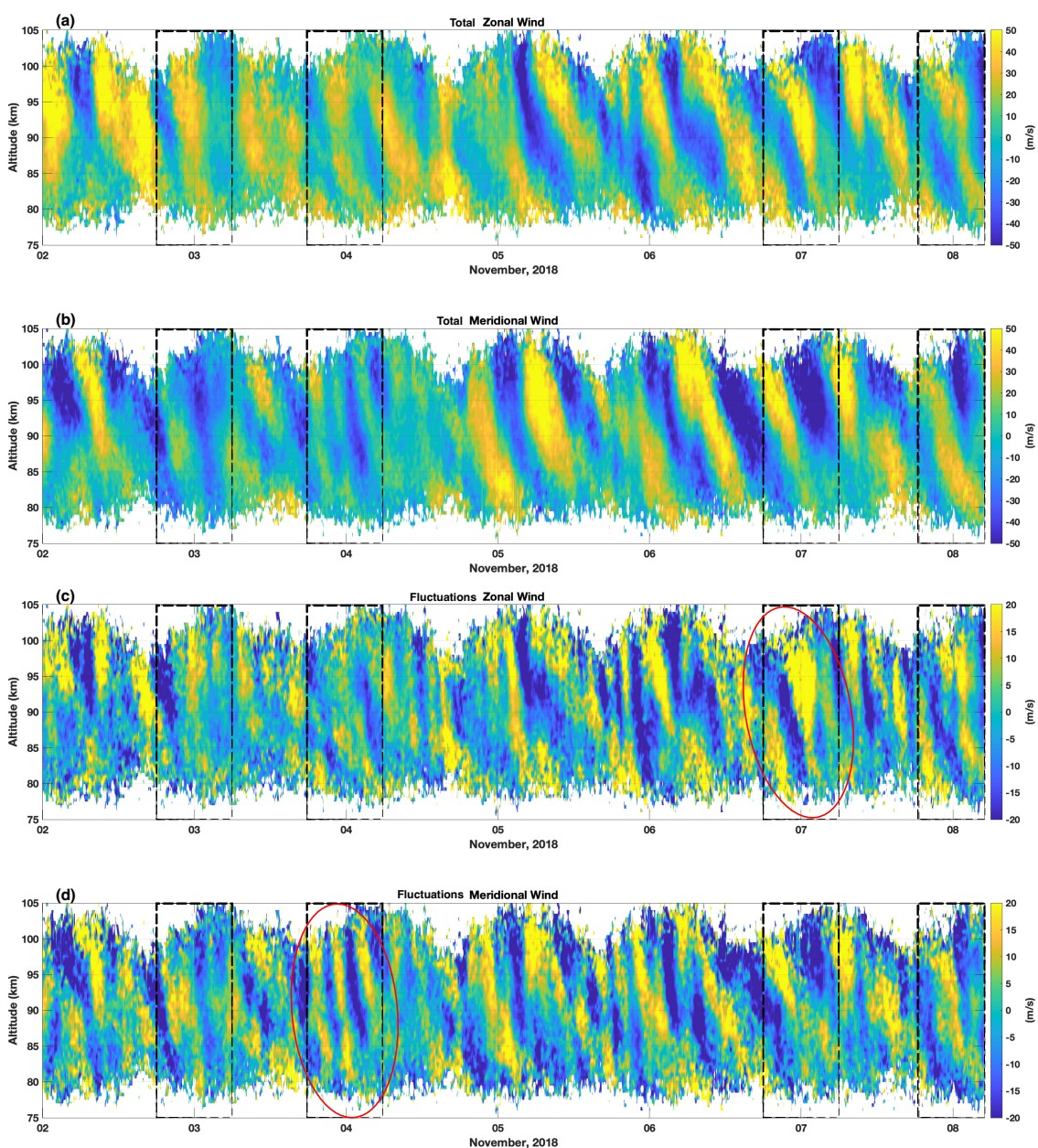

**Figure 2.** (a) Zonal and (b) meridional wind measurements for the duration of the SIMONe–2018 campaign generated by the MSMR network. Note the dominance of the semidiurnal tide on the horizontal wind. (c) Zonal and (d) and meridional wind fluctuations of $\tau_o \leq 4$ hours. Note the presence of coherent gravity wave features (red ellipses) on Nov. 3–4 in the meridional wind fluctuations and on Nov. 6–7 in the zonal wind fluctuations coincident with enhanced keogram brightness for the same nights in Figure 1. Dashed boxes indicate hours of simultaneous operation of the ASI and MSMR systems.

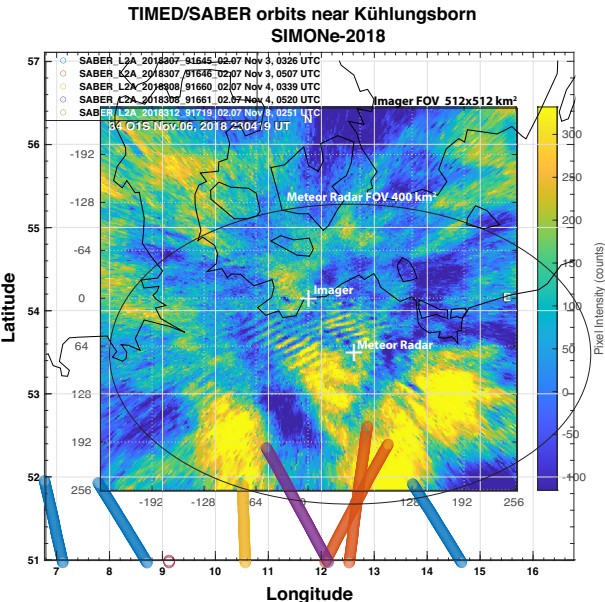

**Figure 3.** Individual TIMED/SABER satellite orbits near Kühlungsborn during the SIMONe–2018 campaign. The colored lines represent the location where vertical atmospheric profiles were measure, not the actual satellite locus. The day and time of each orbit is indicated in the legend. The field of view of 512x512 km$^2$ of the airglow camera projected at $\sim$ 95 km is indicated by the O($^1$S), TD image mapped onto geographic coordinates, while the ellipse indicates the field of view of the MSMR system. The white crosses indicates the coordinates of the imager system in the Kühlungsborn observatory, and the meteor radar system.

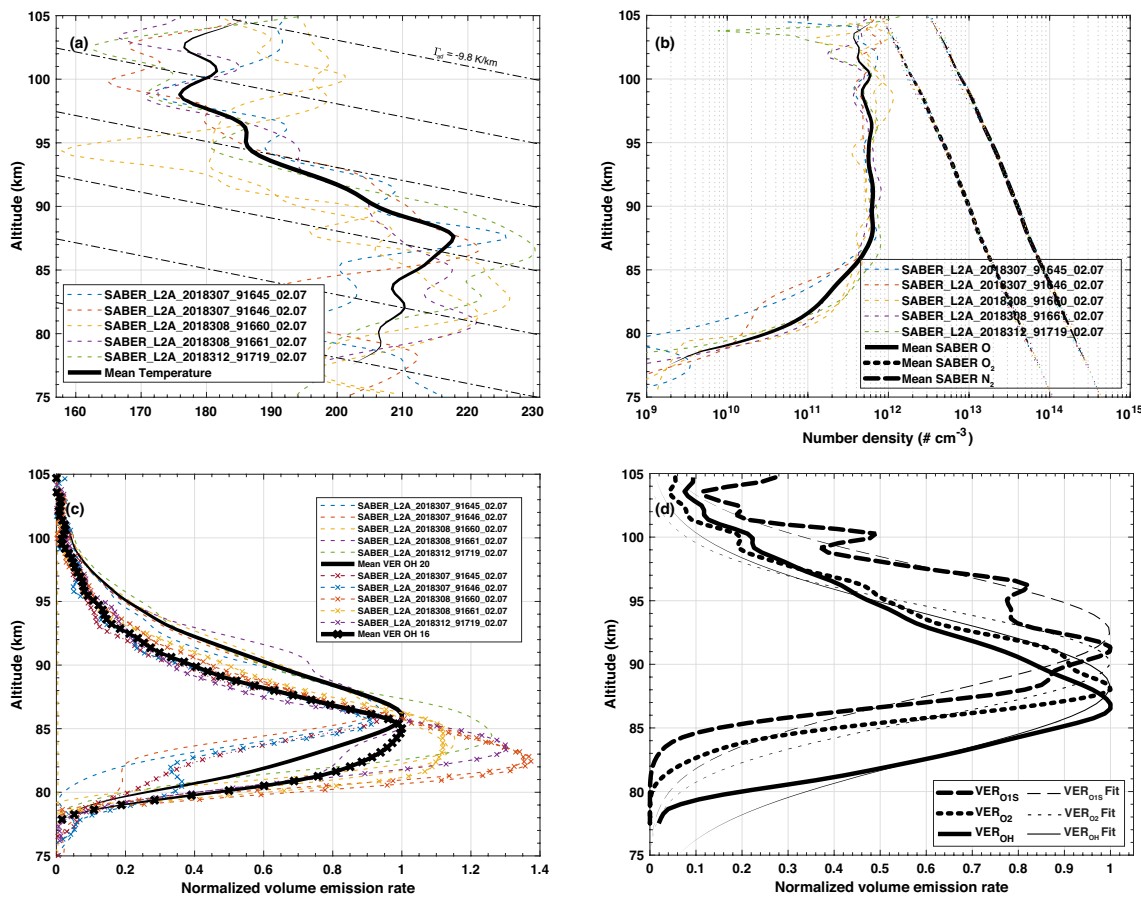

**Figure 4.** (a) Temperature, (b) atomic oxygen, molecular oxygen, molecular nitrogen number densities, and (c) OH20 (2.1 $\mu$m OH(A)) and OH16 (1.6 $\mu$m OH(B)) volume emission rates collected by TIMED/SABER satellite near Kühlungsborn during the SIMONe–2018 campaign within four degrees of latitude or longitude of the observatory. (d) Calculated volume emission rates for OH(8,3), $O_2(0,1)$, and $O(^1S)$ layers (thick black lines) using SABER mean profiles in panels (a) and (b). Colored dotted lines in (a), (b), and (c) indicate individual orbits of the satellite, while thick lines indicate the mean of the individual orbits. Gray dash-dot lines in (a) indicate the adiabatic lapse rate $\Gamma_{ad}$ =-9.8 K/km. Thinner black lines in (d) are Gaussian fits of the calculated VER profiles. Individual VER airglow layer features for both measured and calculated VER are in Table 2. We have used SABER data version 2.07 to compose these plots.

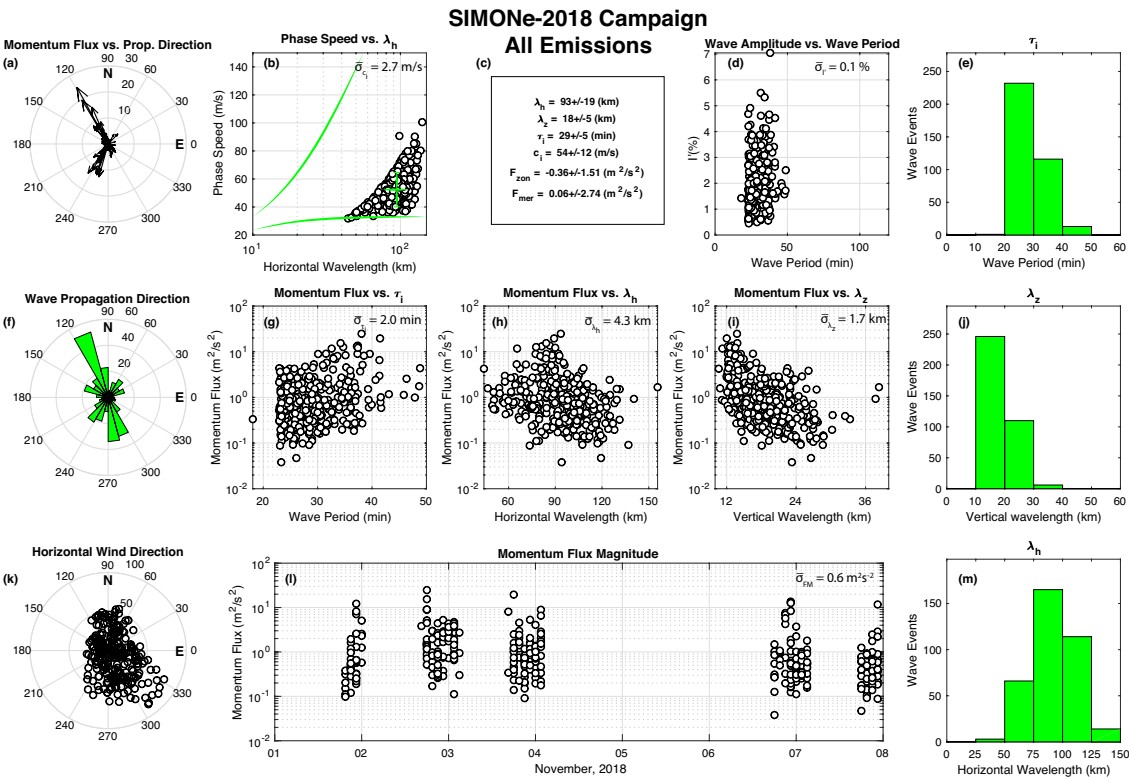

**Figure 5.** Short-period wave parameters obtained from OH, $O_2$, $O(^1S)$, and Na airglow image analysis using the auto-detection method (Tang et al., 2005; Vargas et al., 2009; Vargas, 2019). The mean measurement error is indicated in the chart of each wave parameter. Plots of waves detected in each emission separately are in the supplementary files of this paper.

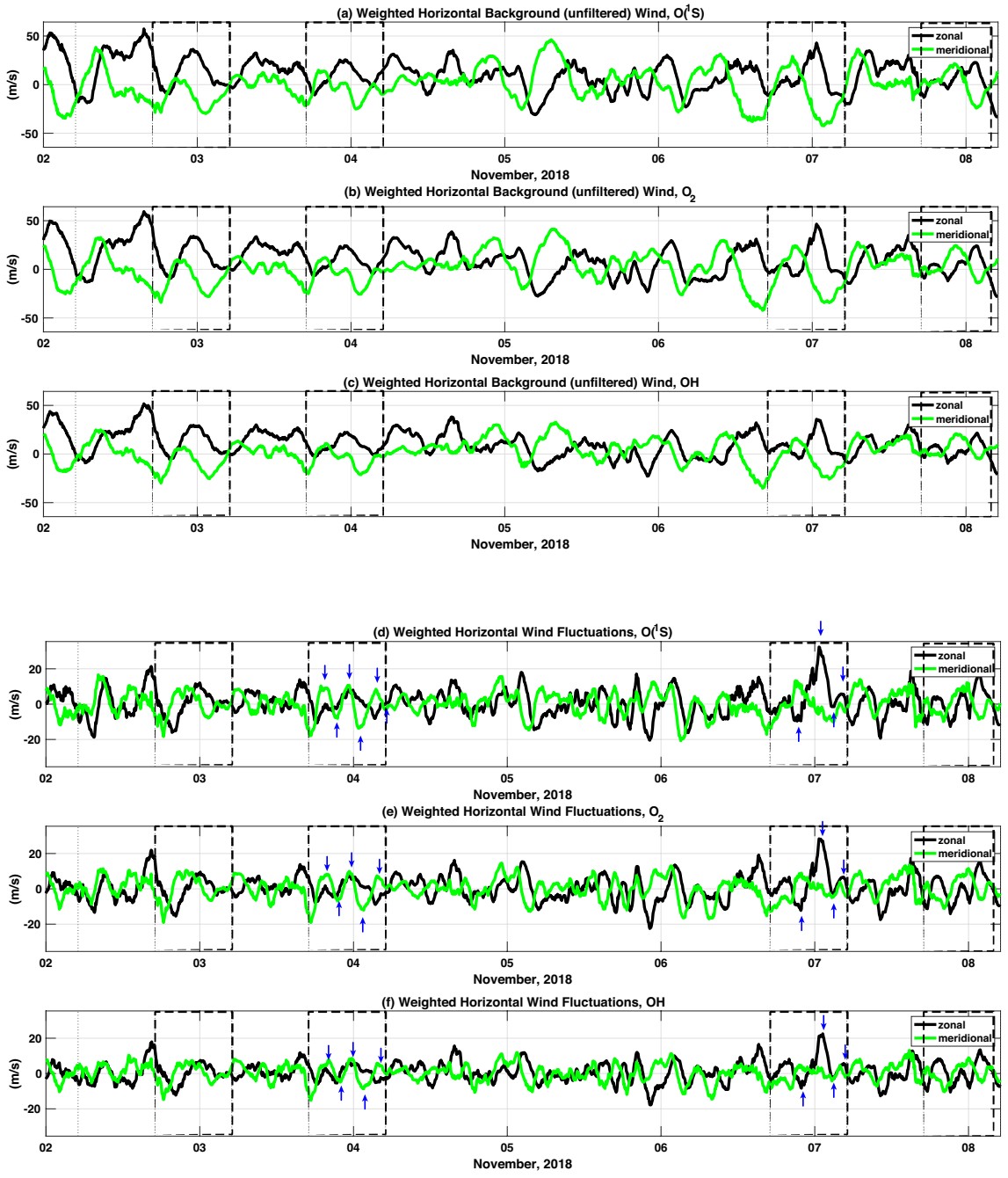

**Figure 6.** (a) O($^1$S), (b) O$_2$, and (c) OH volume emission rate weighted zonal and meridional background (unfiltered) winds. (d) O($^1$S), (e) O$_2$, and (f) OH volume emission rate weighted zonal and meridional wind fluctuations. Wind fluctuations were obtained first by averaging the wind over 400 km$^2$ field of view using a 4-hour temporal, 4-km vertical windows to obtain winds representing large-scales variations, then subtracting these estimates from the background wind field. The vertical blue arrows indicate coherent wind fluctuations also seen in the airglow brightness. Dashed boxes indicate hours of simultaneous operation of the ASI and MSMR system.

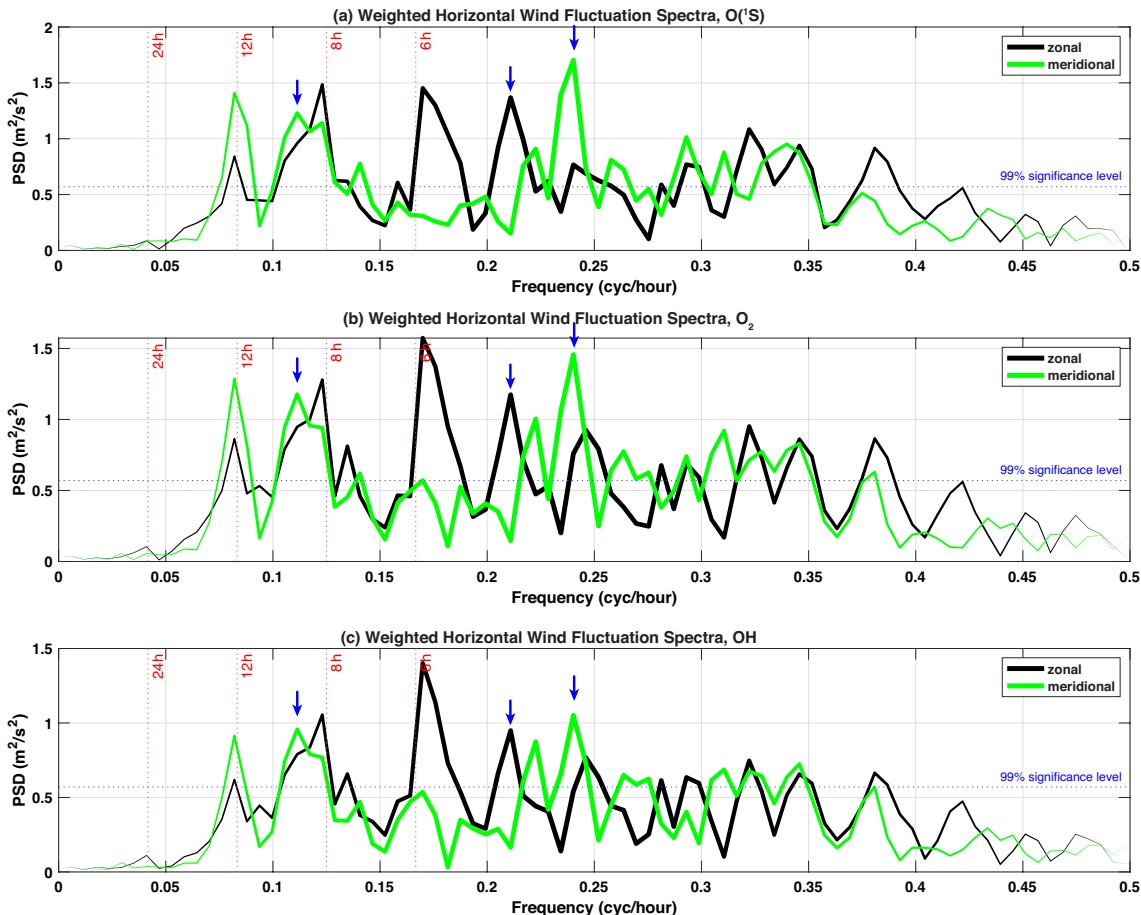

**Figure 7.** Spectra of weighted zonal and meridional wind fluctuation of Fig. 6. The dashed red lines indicate tidal periods. The vertical blue arrows indicate wave frequencies of persisting wave structures also seen in the airglow brightness. Statistical 99% significance level is indicated by dotted blue lines.

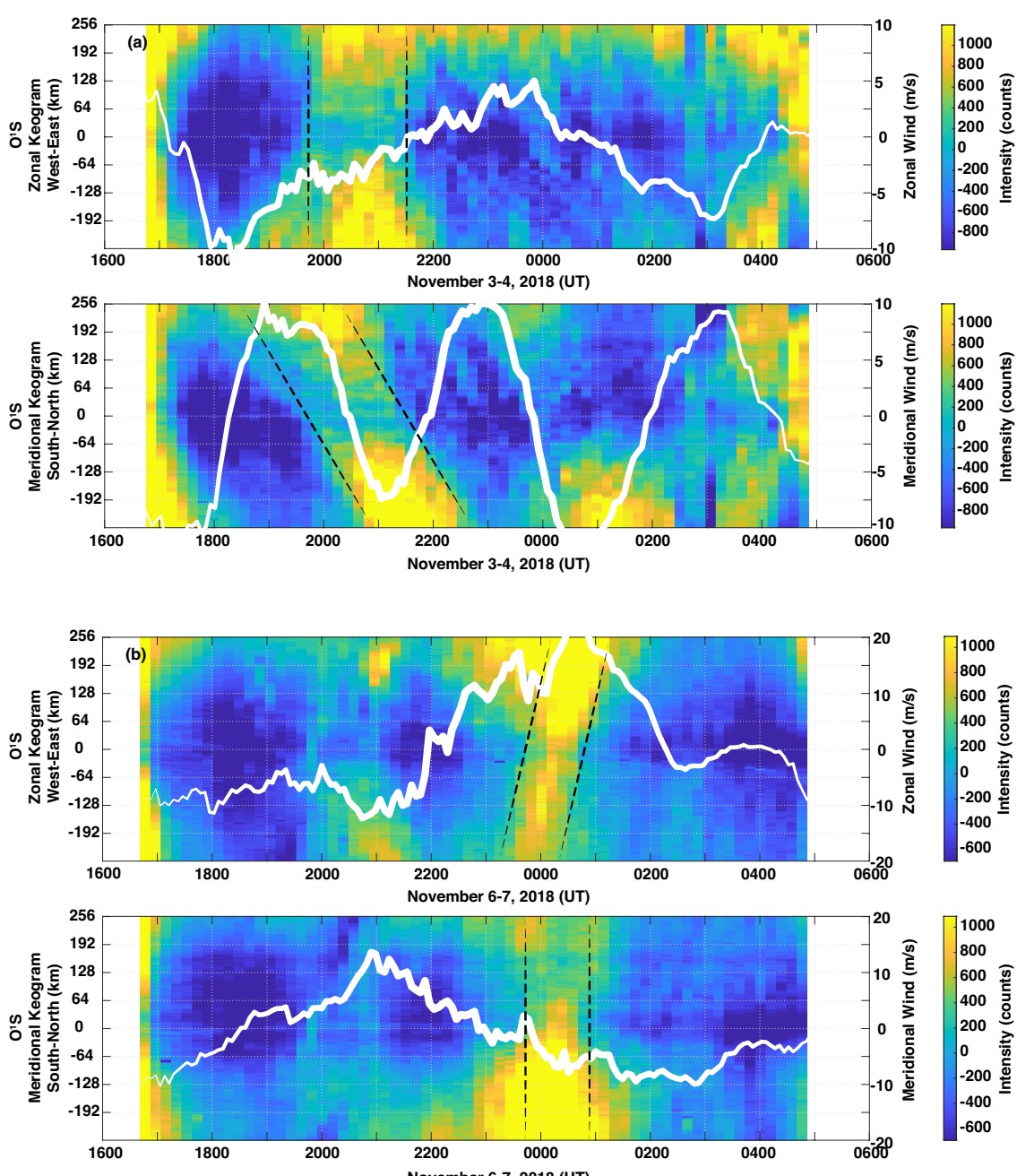

**Figure 8.** Enhanced contrast keograms of O($^1$S) airglow for (a) Nov. 3–4 and (b) Nov. 6–7, 2018. The keogram of Nov. 3–4 shows a large amplitude, large-scale gravity wave at 2000-2200 UTC heading south. A large-scale wave is also seen on Nov. 6–7 propagating eastward at 0000 UTC. The white continuous lines on the keograms indicate the wind fluctuations weighted by the O($^1$S) volume emission rate.

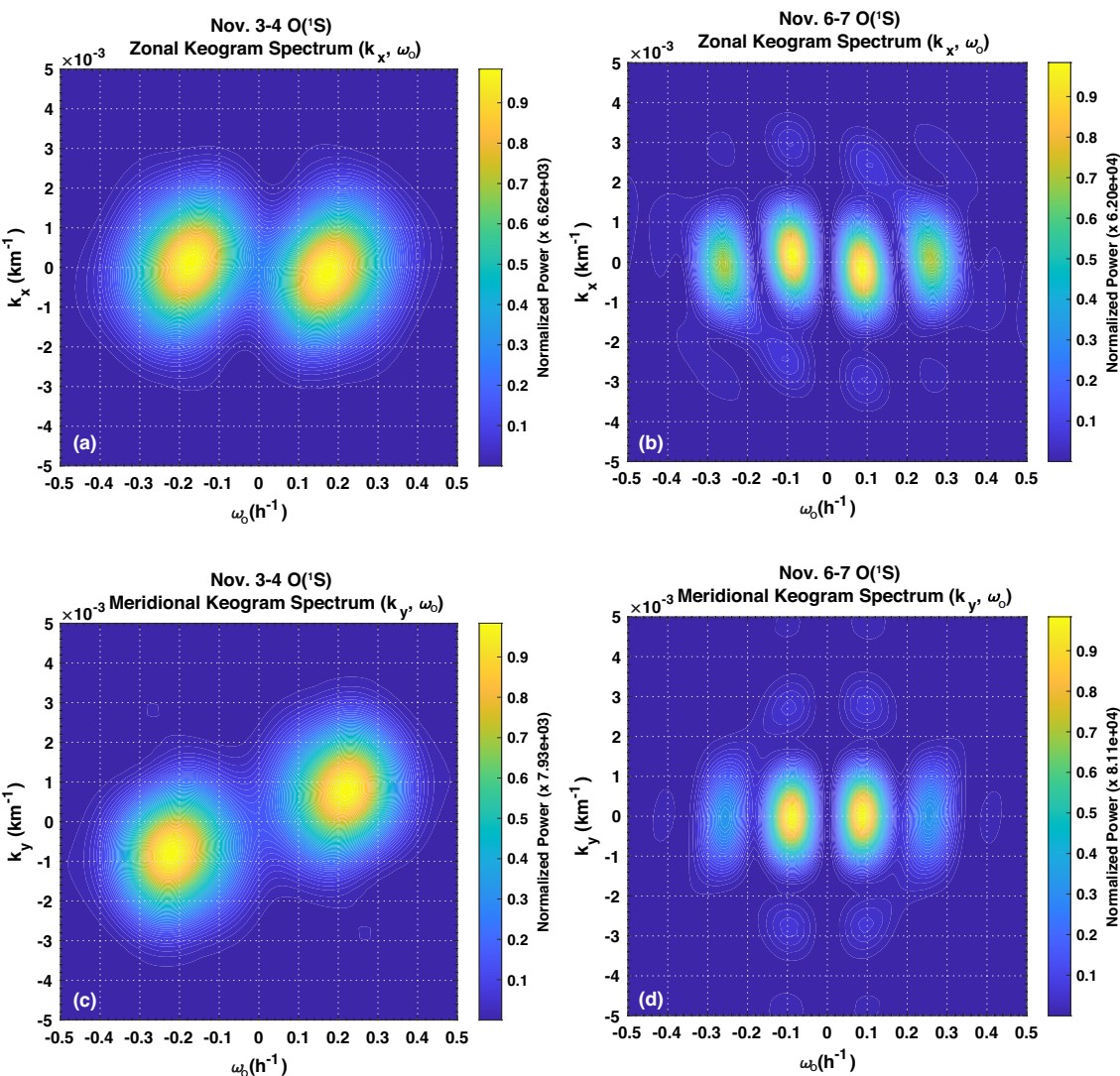

**Figure 9.** Composite $(k_x, \omega_o)$ and $(k_y, \omega_o)$ spectra of the keograms in Fig. 8 for the nights of Nov. 3–4 (panels a and c) and Nov. 6–7 (panels b and d).

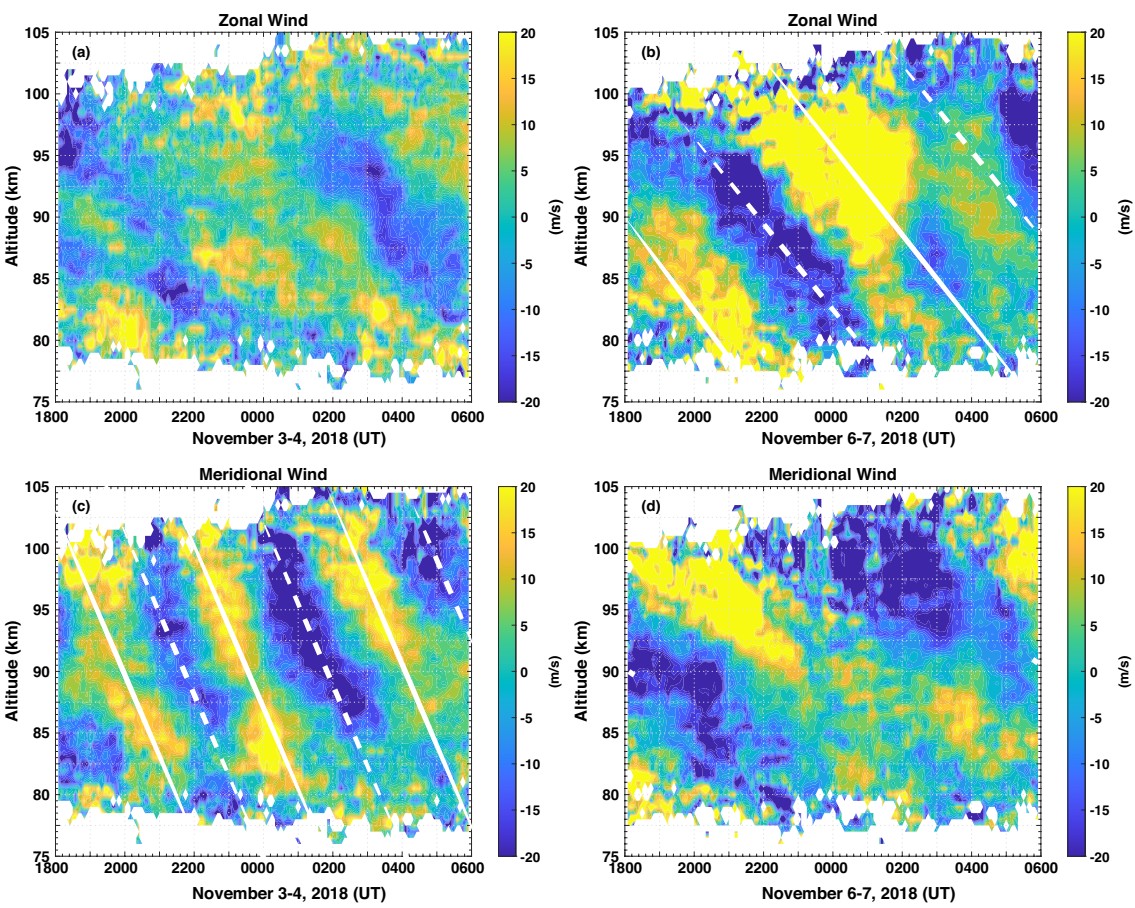

**Figure 10.** Time-altitude cross section of the zonal and meridional wind fluctuations for the nights of Nov. 3–4 and Nov. 6–7, 2018. The continuous (dashed) white lines indicate crests (troughs) of the oscillations as well as the descending phase (ascending energy propagation) of the waves. Notice the coherent $\tau_o \sim 4.3$ hours gravity wave oscillation on Nov. 3–4 with $\lambda_z \sim 25.6$ km. The zonal wind oscillation on Nov. 6–7 corresponds to a $\lambda_z \sim 21.3$ km, $\tau_o \sim 8.0$ hours gravity wave.

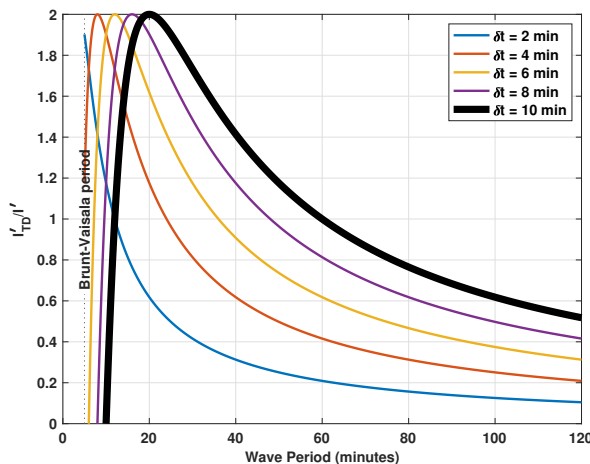

**Figure 11.** Effect of time difference image filter on gravity wave amplitudes of various periods and acquisition times ($\delta t$). The black thick line corresponds to the acquisition time of our imager during SIMONe–2018.