# Peer review of "Mesospheric gravity wave activity estimated via airglow imagery, multistatic meteor radar, and SABER data taken during the SIMONe-2018 campaign"

_Atmospheric Chemistry and Physics, 2020_

## Referee Comment (RC1) · Anonymous Referee #1 · 5 Oct 2020

This manuscript presents a fairly detailed investigation of gravity waves observed in airglow image data obtained during an observation campaign in November 2018, from Northern Germany. It is well-written and provides a good assessment of the effects of GW on the mesosphere, even if it is during a limited period of time. However, the authors should address the following comments:

- It is a little bit surprising that there are no waves with intrinsic period <20 min. Most of the waves should be propagating against the wind, therefore their intrinsic phase speed should be larger than the observed one, and their intrinsic period should be

smaller than their observed one. - lines 255-271: It is complicated to compare the results with Li et al., 2011, because of the time resolution (∼10 min) which does not allow for detecting most short-wavelength, short-period waves. As also noticed in the manuscript, this campaign covered only a few days, while Li et al. results represent the whole year. Same problems with Li, 2011. - lines 284-285: These mean values are very small, especially considering the error. What does -0.36+/-1.51 mean? - 0.36+1.51=1.15, so the MF is carried in the other direction? Or you are sure about the wave direction of propagation, and so the minimum possible value is 0? I am not sure how it is possible to measure such small MF values giving the uncertainties in the measurements. - line 286: The total value is very misleading and cannot be used for any future comparisons. With the analysis method used in this paper, GWs are measured for each image, therefore the more images, the larger the total MF. If the imager cycle was 1 min instead of 10 min, the total MF would be 10x larger because 10x more waves parameters would have been measured! An average MF value for each wave and its duration could be more useful to assess its impact. The statistical results (11% of the waves responsible for carrying 50% of the MF...) are still relevant but the authors have to be very careful when giving the "total" MF. - The number of waves detected (362) is also controversial. There were NOT 362 separate GWs propagating over the observation site during these 4 nights! There were 362 wave measurements, which is different. Some of the waves probably lasted for hours and their parameters were measured several times. Of course, these parameters can evolve depending on the forcing and the background atmosphere conditions, but there were still the same GWs coming from the same sources. - The GWs were observed in 4 different airglow layers, at 4 different altitudes. Could you give more information about that? Which waves were observed simultaneously? It would also be interesting to see the difference in MF between the airglow layers. The MF divergence would be very interesting, as well. - You should mention ducted waves. Even if the waves presented in this paper are unlikely to be ducted because of the minimum intrisic periods, these waves exist but do not carry any vertical MF, so they would biased your results. Again, a comparison

between the different airglow layers would help. - It is probably beyond the scope of this paper, but given the small number of nights, looking for sources (tropospheric weather, convection, fronts...) would be interesting.

Minor comments: line 12: can you add the number of nights/hours of observation? It would give a better idea of the importance of the MF values. line 26: remove "successfully" line 93: long-period variations Section 2.2: what are the vertical and temporal resolutions of the radar system? line 137: FWHM line 175: "duration" instead of "length" line 177: The MF equation is proportional to Lz/Lx, so it is not that surprising that large horizontal wavelength waves carry less MF. It is more surprising that large vertical wavelength waves carry less MF, though. line 191: You must have done your filtering wrong if the cutoff was 5 hours and you still see that many peaks, especially the ones around 6, 8, and 12 hours! The peak at 8.9 hr is pretty close to the peak at 8 hr. You have to give the resolution of the spectra, otherwise this is not very convincing. line 204: The keograms could be improved by flat fielding the images first. Not sure if this had been done, but it would improve the signal at the zenith compared to the edges. Figure 8a bottom is misleading. It looks like the wave tilt (which is related to the direction of propagation and phase speed) corresponds to the decrease in meridional wind intensity. Not sure how to avoid that. You should add the directions of propagation in Table 3. line 221: What is the error on the horizontal wavelength measurements? line 235: 8 hours, could be a tidal component. line 293-294: The last sentence is enigmatic. If these larger amplitude waves are only seen in the O2 emission but not below (OH, Na) or above (OI), it is quite puzzling. Lines 363-365: Can you rephrase this sentence? It is not very clear. Line 378: The links are missing! Figure 4: (d) Calculated volume emission rates... Figure 7: Can you add a detection threshold?

The comparisons with previous publications are lacking some relevant studies: eg, Bossert et al., 2015, for MF of long period waves, Cao and Liu, 2016, for the impact of large vs small MF GWs, and even on the same topic Hertzog at al., 2012, Plougoven et al., 2013, or Wright et al., 2013. The last 3 describing stratospheric measurements,

though. The comparisons with Li, 2011, and Li et al., 2011, are not very relevant except because they used a similar analysis method.

---

## Referee Comment (RC2) · Anonymous Referee #2 · 15 Oct 2020

This manuscript deals with gravity waves detected using an airglow imager and discusses their effect on the background wind, combining collocating meteor radar network data and satellite data as background information. Their effort to make full use of the obtained data during the limited campaign period is much appreciated: expanding their analyses to waves with horizontal scales larger than the field of view of the imager. However, the overall analysis methods often lack detailed explanation and hard to follow, and some part of them even seem inadequately done. The authors are advised to revise the manuscript substantially by addressing the following comments.

[Figure]

Major comments:

The sequence interval of the imager observation is 10 min, and the exposure time is mostly 2 min. This means that possible aliasing should be taken into account in the analyses although it is not described in the manuscript. Because of the 2 min exposure, any structure with a period shorter than 4 min will be smoothed out and not be resolved, but those with a period between 4-20 min can easily alias into slower period motions, and significantly affect the results.

In detecting wave structures from wind data, a high-pass filter with a cut-off of 5 hr is used. However, the detected wave periods are around 4 and 8 hr, which are close to or longer than the cut-off. The wave amplitudes and also phase structures will be largely affected. Even a spectral analysis is done using the filtered wind, and fluctuation components with periods longer than 4 hr are discussed (Fig 7), leading to the subsequent analyses of those components based on the filtered wind as shown in Figures 8 and 10. Although the authors mention these peaks as 'leakage' (line 190), I am afraid that such data treatment is not acceptable at all. What the authors can remove before the spectral analysis in the present case would be only the background wind (mean wind) and trend. Since the imager data is unfiltered, wind values should be treated in a similar manner in comparison.

The details of 2D spectral analysis method shown in Fig 9 are not clear. The keograms shown in Figure 8 are thought to be used, of course, but how is the calculation done? The borders of the keograms can significantly affect the results if miss-handled. Isn't there a possibility that the evening twilight seen around 5 pm affects the results? In other words, what kind of spectral window is applied in the analysis?

Minor comments:

Abstract

Line 1 'large and small'

Not quantitative. More specific description is wanted. The same comments apply to the other 'small' and 'large' in the manuscript.

Line 6 'flux divergence estimations'

Momentum flux is estimated, but the divergence does not seem estimated, instead MF is just assumed to be deposited around the mesopause.

Line 11 Mean MF, total MF and number of detected events

585.96/362 is not 0.88. How is the mean estimated? Is the detected number 362 the number of independent wave events, or multi-counted number? The same comment to the lines 285-286.

Lines 17-18

The lidar data in the upper stratosphere is not actually analysed in the manuscript, but only qualitatively mentioned. This statement does not seem appropriate in the abstract.

The main body of the manuscript

Line 52 'modes'

Meaning not clear.

Line 58 'large amplitude filtered wind fluctuations'

Meaning not clear.

Line 84 'long period oscillations'

The wave structure is not very clear. Even one cycle is hard to distinguish, especially for the Nov. 06-07 case.

Line 91 'time difference'

Is this done using pairs of the adjacent 10 min separated images? The detail should be written.

Line 116 'smaller' > shorter

Line 116 'simultaneously the back…..'

'with' after 'simultaneously'

Line 118 'Fig. 1c'

Meant to be Fig. 1d?

Section 3

The definition of 'small' and 'large' should be described at the beginning of Section 3.

Sub-section 3.2

The meaning of 'fluctuation' at line 182 is not clear although it is inferred later at lines 190-191. It is confusing. The filter characteristics also should be written in the caption of Figure 6.

Lines 210, 211, 219, 220, 224, 234, 235 316, 317, 320 and 321

Why are the uncertainly of all these values '+-1'?

Line 244 'amplitude'

Since the background wind is not thought to be a wave, rewording will be appropriate; something as magnitude.

Line 245

Absorption occurs at a critical level, but reflection occurs at a turning level. Different wind structures are responsible, perhaps as you know.

Lines 283-284 'toward the west of -0.36+-1.51 m2/s2'

Very confusing description.

Lines 286-288

The finding of a small number of waves carrying a large portion of MF is a good point. I suggest that the authors include discussion on GW intermittency and its importance in model atmosphere studies.

Lines 289-294

This can largely affect the obtained results. Hopefully more discussion wanted. Isn't there any possibility that the layer thickness is related, or is it already taken into account in the analysis? It seems from Figure 4 that O2 layer is the thinnest.

Line 296

Yes, if the acceleration or deceleration occurs all along the same latitude circle, to my understanding.

Lines 339-340

I think this is a nice point. In the radar studies, a long-lasting topic has been which period range of GW is most responsible for MF deposition. Short period GWs were thought to be responsible before, but recent studies suggest that longer period waves are responsible. One example I know is Sato et al, 2017 (https://doi.org/10.1002/2016JD025834). There can be more. Including such preceding studies will be beneficial for even more fruitful discussion.

Figure 2 '> 5 hours' should be '< 5 hours'?

Figure 5

Are these dots independent wave events with each other?

Figure 6

Meaning of 'fluctuation' should be described clearly. High pass filtered?

Figure 7

I am afraid that a spectral analysis after applying a filtering is not acceptable.

Figure 8

Image data and wind data should be treated in a similar manner; detrend, background removal or whatever.

Figure 9

The way of 2D spectral analysis should be described in the manuscript.

Figure 10

If all the figures are processed with the high pass filter with the cutoff of 5 hr, reprocessing is suggested, at least, for the November 6-7 case.
* * *

---

## Short Comment (SC1) · 15 Oct 2020

We would like to thank the reviewer for taking the time to review our work. Your help is deeply appreciated.

A short reply to major issues is given here and is to be elaborate timely. -> as pointed out, we did not detect waves with periods <20 min, although they would be present certainly. Due to the filter wheel cycle time, a given airglow emission is sampled every 10 min, which allows only the detection of waves having periods >20.

[Figure]

-> compare the results with Li et al., 2011. we did the comparison because the paper utilizes the same autodetection technique. we focus on the results of Li et al which are comparable to the wave scales we have detected. We will check this to provide a stronger correspondence btw the results.

-> lines 284-285. the meaning is relative to the MF mean and standard deviation of the sample, not the error in the MF determination.

- line 286: I agree the total MF value means nothing.

-> momentum flux of waves in different layers can be given. we have estimated it separately, but because the detections in an individual layer were not too many, we showed the results together.

-> estimating the momentum flux divergence between two layers is possible and we have done that in the past (Vargas et al, 2015, http://dx.doi.org/10.1016/j.asr.2015.07.040). The simultaneous identification of a wave in two layers is more demanding, and we do that for specific, individual cases. Instead of that, we have estimated the flux divergence from our model (Vargas et al, 2007, JOURNAL OF GEOPHYSICAL RESEARCH, VOL. 112, D14102, doi:10.1029/2006JD007642, 2007) based on the distribution of periods and wavelengths of the observed waves. In this paper, we have done that as if the wave amplitude was measured in the OH and O(1S) layers simultaneously, with no change in amplitude detected (saturated waves).

-> we did not address the sources of the waves in this paper but will do it in a following publication exploring the sources of primary and secondary waves based on secondary wave generation in the range of 30-40 km as seen in lidar temperature data.

---

## Referee Comment (RC3) · Anonymous Referee #3 · 23 Oct 2020

**General comment:**

The authors describe different measurements they carried out during the SIMONe-campaign in November 2018 at or around Kühlungsborn, Germany. Supplementing satellite-based data are used. They combine the measurements in order to fully characterize gravity waves and discriminate between large and small-scale waves. However, they do not explain what they understand under small and large scale.

From my point of view, there are two main results. The authors found out that approximately 11% or all detected waves carry ca. 50% of the total momentum flux and they had a specific look on two large wave events. They motivate that these structures might be due to secondary gravity waves. Nevertheless, this remains speculative.

The manuscript is well-structured and sections 1 to 4 are easy to read in the first approach. However, on closer inspection important information is missing. In section 5, the formulas disturbed my flow of reading. Following the classical approach, they should appear earlier in the manuscript which would help here. The data and the methods used to measure and identify the gravity are appropriate, however especially the data section could have some more details. My main point concerning the section results refers to error bars and/or significance levels: I didn't find a single one.

I propose to accept the manuscript after some major revision.

**Major points**

The manuscript is written in a rather abbreviated style.

The most important point in this context is that the manuscript suffers from a lack of information about the quality of data and analyses. I found neither a single error bar nor a significance test in the section results (in the discussion part there are very few error bars). For clarity issues, it is probably not helpful to provide this information directly in all figures (for example I have no idea how to incorporate error bars in keograms), but it can be done in some cases (FFT) and in the others the information can be given in the text.

Parts of the main results depend on statistical analyses. A second point in the context here is that the authors do not explain how they define wave event and which consequences this definition has on their results. If I operate an imaging system which takes a picture every ten minutes, is each result of the spectral analysis of each image defined as a wave event or is a wave event an oscillation in space which shows nearly the same horizontal wavelengths over some time (so in some images). Fast waves would be underrepresented in the first case. Does this have any influence on the results?

Another point is: It is really strange to see a paper where TIMED-SABER data are used and Martin Mlynczack and James Russell (or at least the SABER team in general) are neither co-author nor mentioned in the acknowledgements. There isn't even a single citation of them (e.g. Russell et al., 1999, https://doi.org/10.1117/12.366382 or Mlynczak, 1997, https://doi.org/10.1016/S0273-1177(97)00769-2). On the other hand, there exist co-authors here who also "only" delivered data. This is not consistent. Maybe the authors offered co-authorship to the SABER people and they declined, I

don't know that, but is there really nothing to mention in the literature list or in the acknowledgements?

**Minor points**

**Section 1**

General comment: The introduction is quite general. In my opinion, the technical focus of this manuscript is on the combination of different measurements to describe gravity waves as comprehensively as possible. Airglow imagers are used in order to derive horizontal wavelengths and periods. Wind information come from radar data. Temperature information are based on SABER data. Other groups have already done such studies or similar ones - just enter 'airglow radar SABER gravity waves' in google scholar. However, there is no reference in your manuscript. One of the main topics of the manuscript is the derivation of the momentum flux. There are already other publications on this topic which are not mentioned here or in the discussion (e.g., Ern et al., 2011, https://doi.org/10.1029/2011JD015821 derived the momentum flux globally based on satellite data). Please include some citations and put your work into the context.

p. 2 l. 48 Can you please provide the exact campaigns period?

p.3 l. 55 & 57 What is small and large scale for you? This comments also refers to the headings of section 3.1 and 3.2.

**Section 2**

From my point of view, the level of detail of the different subsection varies (2.2 has more details than the two other subsections). Additionally, I do not get all "classical" information for all instruments such as spatial and temporal resolution or some information about the data quality (not necessarily bias and precision but for example comparisons with other instruments, for SABER such studies are available).

I also miss further literature for ASI and SABER where the reader can look up (technical and data processing) details about the instruments (e.g. concerning ASI the detector size etc., concerning SABER the retrieval etc.).

For SABER, different data versions exist. I conclude from the legend of figure 4 that you use the newest one (2.07) but not every reader might be familiar with the "SABER notation", so please provide this information also in section 2.3.

p. 4 l. 87 In the text you use UTC, in figure 1 you use p.m. If it was consistent it would be easier to read.

p. 4 l. 94 You write that the result of a time-difference operation is an image where the contrast of small-period, small-scale oscillations is enhanced. I think with small-scale you mean the spatial dimension. In this case, I do not fully agree with you. Whether you enhance a signature or not should depends on the speed of the signature: if it is near zero you won't see it in your time-difference image

because you just subtract it. The phase speed is directly proportional the wavelength (in this case the horizontal one) and indirectly proportional to the wave period. The phase speed significantly differs from zero, the larger the wavelength or / and the smaller the period is. So, you enhance small-period but large-scale oscillations.

p. 4 l. 115 Which high-pass filter exactly? If you say you use a five-hour high pass filter, it means for my that all periods longer than five hours can pass. Later in the document, it becomes clear that the opposite is true and you probably referred with the term "high-pass" to the frequency domain. Can you please clarify here which periods are still present in the filtered data. Additionally, in the spectra in figure 7 you still see a rather dominant p. 4 l. 118 Are you sure that you really see exactly this oscillation?

p. 5 l. 120 The abbreviation SABER is not explained in the manuscript.

p.5 l. 127 Please concretize the lapse rate you mean.

p. 5 l. 127 Please be a little bit more precise here. It is not important that the orbit of TIMED is exactly over the observatory, the field of view of the instrument has to measure over or near the observatory.

p. 5 l. 128 From figure 3 it looks like you use also profiles which were not within the field of view of the imager.

p. 5 l. 133 It is great that you are able to calculate the VER profiles for the different species but why don't you use the ones provided by SABER?

p.5 l 137 Please correct the letter shift (FWMH → FWHM) and explain the abbreviation.

**Section 3.1**

The way to present such campaign results is always a tightrope walk. If the reader is flooded with details, he loses the overview, if it is too short, questions remain open. For me questions remain open.

I miss two formulas (dispersion relation for high-frequency waves and vertical flux of horizontal momentum, or at least citations of literature where the reader can look them up) and in the case of the dispersion relation I would like to read some information about the Brunt-Väsälä frequency (probably calculated based on SABER, VER weighted?).

Please also provide the information that the cross-spectrum is the Fourier transform of the cross-covariance function, since you treat pictures you probably use the 2D version. Another useful information would be the sensitivity of the analysis. You calculate the cross-covariance of two TD images. The calculation of TD images already means filtering the data for fast waves. The calculation of the cross-covariance provides you information about the waves which change from one TD image to the other. So, there are two stages of filtering and it would be interesting to provide information which waves pass this filter and which don't.

p. 6 l. 165: I think you mean 5j instead of 5e and the wavelengths range between 10 km and 40 km.

p. 6 l. 169: I see 100 m/s instead of 80 m/s.

p. 6 l. 176: "larger momentum waves on Nov. 2-3 and Nov. 3-4." Ok, it's a log plot but the maxima of these two nights are not so much higher than the maximum of Nov. 1-2, for example, which brings me to the question of error bars. If you plot them, figure 5l might become a little bit busy, so this might not be the optimal solution, but in any case you should give information about the error bars somewhere (also concerning the other parameters shown in figure 5) and state whether the sentence about the date of appearance of the larger momentum waves is still valid in this case.

**Section 3.2**

p. 7 l. 184 "once the layer peaks within +/-2 km from each other"? The wind fluctuations weighted with the VER of $O(^1S)$ and OH(8-3), so figure 6 d and f, look really similar, even though the respective peak heights are more than 4 km away from each other.

p. 7 l. 185 The citation you provide refer to the equatorial region, you should mention at least one citation referring to mid latitudes (e.g., Wüst et al., 2017, https://doi.org/10.5194/amt-10-4895-2017). In any case, 15 km is indeed large. When I look at figure 4 c (VER measured by SABER), I find values of 9 km to 11 km which is much more in the range of other authors. Your simulated profiles in figure 4d (thick lines) show higher values but none of them reaches 15 km. So, first I wonder where the 15 km FWHM comes from and then I would be interested in why is there a difference between simulated and measured values? Which one can I believe or are both ok within certain limitations? And now we are again in the range of error bars, quality of the data etc.

p. 7 l. 187 The values after the +/- are what?

p. 7 l. 189 I miss a significance level in figure 7.

p. 7 l. 193 Please omit "salient" – this peak is as salient as the 12 h peak for example, which is only due to leakage as the authors say. By the way, I find it strange that peaks which are due to leakage are as prominent as others. Here, a significance level and more information about the filter as already mentioned before would help.

p. 7 l. 211 I find it difficult to see a period of 8 h±1 h.

p.7 l. 215 Which spectral analysis? Please concretize.

p. 7 l 217 The keogram analysis does not only provide horizontal wave numbers, you also mention the wave frequency in figure 9. Can you please concretize this in the text? Furthermore, you mention the observed frequency $\widehat{\omega}$ in the text but the frequency $\omega$ in figure 9. You probably mean the observed frequency in figure 9, don't you?

p. 8 l. 118 &v226: Is there a difference between ~0 and $0.0\times10^{-3}$ within the unknown accuracy of your analysis and the unmentioned significance levels?

p. 8 l.228 & 230: You mention that the phase is propagating downward but the wave is propagating upward? Don't you mean the energy is propagating upward?

**Section 4**

p. 8 first paragraph: you argue that the atmosphere is convectively stable using the mean temperature profiles over some days. When discussing dynamic instability you look at individual measurements and argue that they can reach 50 m/s and more. This is not consistent having the mean on one side and the individual profiles on the other. Please go a little more into detail here. Furthermore, you write that since the wind field is rather strong it could cause critical levels – wind fields can always cause critical levels. It just depends on the phase velocity of the wave. You probably mean that large wind speeds are more likely to cause critical levels.

p. 8 second paragraph: You write that the wind is controlling the propagation of southeastward waves via dynamical filtering and draw this conclusion from the comparison of two figures, one addressing the momentum flux direction (figure 5a) and the other showing the wind direction (figure 5k). Why don't you conclude this from the comparison of figure 5f and k? Figure 5a tells you only that there is not much momentum transported in southeastward direction but this can be due to too "weak" waves or too few waves or a mixture of both.

p.9. l. 253 You mention a wind speed of ca. 39.3 m/s. Is this the mean over the background wind or over the total wind (including the gravity wave signatures)?

p. 9 l. 253 & 254: the number behind the plus-minus sign is the standard deviation?

p.9. l. 255-281: you compare your results to two other publications, one of them is a PhD thesis and not peer-reviewed I assume. It is not clear to me why only these two publications are suitable for comparison. There exists a number of other studies since 2011 just type 'airglow gravity waves horizontal wavelength' in google scholar, even if you restrict your search to 2016 to 2020, you still get many results - not all of them might deliver all wave parameters you derived but even in this case there exist some. So, could you please extend your comparison here accordingly?

p.9. l. 256: You write that your results are directly comparable to Li et al. (2011) since they used a similar auto-detection method. However, you say nothing about the sensitivity of the instruments, which is used by those authors, to the different horizontal wavelengths. The sensitivity of an airglow imager should be determined by the field of view and the number of pixels your imager has, for example. As you mention in line 265 also other parameters such as filter wheel cycles or integration time determine the sensitivity. Can you please adapt the formulation in the manuscript accordingly?

p.9 l. 267: Please replace min by minutes

p.9. l. 269: You write that the filter wheel cycle which is used does not only influence the derived periods, it seems to influence also intrinsic phase speeds. It should! Phase speed is related to the ratio of wavelength and period. The filter wheel cycle influences the sensitivity of your measurement system in the temporal domain, it does not affect the spatial domain. If you can detect waves which are characterized by smaller periods but by horizontal wavelengths in the same range than before, you will automatically see faster waves.

p. 9 l. 275-277 Can you please provide citations here or is this information taken out of Li (2011)?

p.9. l. 280 Convective sources during winter? Ok, we could think about strong low-pressure system but since you do not bring further arguments here, can you please mark this sentence as speculation?

p.10 l. 293 & 294 Your vertical wavelength is in the range of the FWHM. According to table 2, the O2 layer is characterized by the smallest FWHM compared to the other layers you observed. So, I would assume the strongest signal in this layer.

p. 10 l. 208 You left out $1/4H^2$ where H is the scale height. Please mention this. Did this omitted factor contribute to the error bar (given in l. 316) or is it not necessary?

p. 10 l. 316 I wonder where the error bar comes from since you didn't provide any error bars in your analysis. Please comment.

p. 11 l. 328 Please substitute m/s by m/s² when writing about g.

**Section 5**

p. 12 l. 362 Where do these numbers come from? I find 22-41 m/s/day in the manuscript and this is for the assumption that the wave breaking continues for 24

**Section Acknowledgements**

p.12 l. 378 Which ftp?

**Figures**

Figure 2

It would be nice to provide the hint that the y-axis of a and b are different to the ones of c and d. Non-radar people might further appreciate a note explaining why there is a daily cycle in the length of the vertical profiles.

Figure 3

This figure is a challenge. I hope I can explain my confusion and I think you can solve most of it by just showing the respective fields of view and not the values measured within the fields of view. First of all: what do the colours in the rectangle and in the ellipse mean, you do not have a colour bar or maybe two? The measurements shown are taken at which day? I think the rectangle shows the imager but isn't the field of view of an all-sky imager a circle or a kind of ellipse when projecting it on a map with an equidistant lat-lon grid? You probably have just cut the area. Then: on the right hand-side of the plot where only the ellipse covers an area (not the rectangle), there is a kind of border. This effect also appears the bottom of the plot. It does not appear where the ellipse ends and only the rectangle covers an area. Why? Maybe you plotted the rectangle over the ellipse. Last: Please insert a y-axis labelling. Puh!

Figure 4

a) Can you please provide the different SABER profiles in different colours or line styles or …? I wanted to find out more about the static stability at individual days but this is not possible if all profiles have the same colour and line style.
Please provide explicitly day of the year and UTC. I can extract it from the legend but this might not be clear for everybody.

c) Can you please scale the x-axis in a way that all profiles also the individual ones are completely within the plot?

Figure 5

a) What do you mean with momentum flux versus propagation direction? If I plot y versus x, I have two axis. You also have two "axis" (one is the radius and one the direction) but the radius seems to show the number of waves or wave events.

c) The value after the +/- is the standard deviation? About how many waves do we talk?

d) j) and m) What is a wave event, so is it the result of one FFT or do you group the results of your FFT, e.g. if a wave is found in one FFT and a wave with rather similar parameters is found in the next FFT, is it the same wave or do you count it twice? Would it make sense to count it twice or would you include a bias in your statistics when counting it twice since fast waves will be counted less often than slow waves?

m) There is a strange white triangle in the lower right corner.

Figure 6

It would be better to extend the y-axis range especially in part (d) to (f) since especially at Nov. 7$^{th}$ the values are not shown in the plot any more.

**Tables**

Table 2

The FWHM I read from figure 4c is roughly 9 km and 11 km for SABER mean VER profiles (thick green and thick black line) – why is there a difference to the values given here?

Here, I come back to one of my earlier comments. You simulate the layer but you also show values measured by SABER. How do I have to judge the differences? For example, you simulate another OH

transition than you measure. The different OH transition dominate at slightly different heights (see e.g. von Savigny et al., 2012, https://doi.org/10.5194/acp-12-8813-2012).

---

## Short Comment (SC2) · 29 Oct 2020

First, I would like to thank the reviewer for taking the time to evaluate our work. I hope to provide satisfactory replies to your comments in a timely manner.

Replying to the major issues found by the reviewer:

We recognize we need to expand the description of the data analysis to make it clear for the readers. Also, we believe the keogram spectral analysis would be understandable as we wrote, but it is obvious we need to specify it in more detail in the manuscript.

[Figure]

Also, we are going also to give deeper thought to the possible aliasing of waves and will try to justify or minimize it. Because every image of a given airglow layer is taken at 10 minutes pace (the imager filter wheel cycle), we will only be able to resolve waves with 20 minutes or longer. However, we believe the aliasing is minimal in this case because the small scale wave's structure (horizontal wavelength less than 500 km) seen in the airglow images are sharp (no smudging).

we will also address the issue with the filtering of the horizontal wind for the wind fluctuations. the filtered wind is used to compare with the large scale waves in the airglow keograms. For correcting the wave period Doppler Shift, we use the total wind at the moment of wave detection, not the filtered wind nor the fluctuation.

---

## Short Comment (SC3) · 29 Oct 2020

**Fabio Vargas**

fvargas@illinois.edu

Received and published: 29 October 2020

We appreciate the reviewer for taking the time to read and make suggestions for the present manuscript.

I understand the reviewer's concern about having written the manuscript in an abbreviate style. Maybe because we wanted to make it concise, we have failed while specifying important parts of the analysis and processing, which must be described precisely to permit the reproduction of the results. Printer-friendly version

The incorporation of error bars in the spectral analysis could be found from our simulations of the error propagation into gravity wave parameters in our recent paper (Uncertainties in gravity wave parameters, momentum fluxes, and flux divergences estimated from multi-layer measurements of mesospheric nightglow layers, https://doi.org/10.1016/j.asr.2018.09.039.)

For the small scale analysis using the autodetection method, we have defined as a gravity wave event the output of the cross-spectrum of two time-difference images when the special peak is larger than 10% of the total energy of the spectrum. two time-difference images are generated from three light frames of the airglow (please take a look at fig 01 for Vargas et al 2009 attached here as supplement, https://angeo.copernicus.org/articles/27/2361/2009/). Thus, if a wave is detected in a given set, it is considered an independent wave detection. Now, if in the next set another wave detection is made, the only way to tell the two events to correspond to the same gravity wave is by comparing their wave parameters. Now, the momentum flux of a wave varies as it goes through the field of view. thus, considering the momentum flux average of all the waves detected during the campaign would be the same as clustering the corresponding wave events into a distinct wave, and then averaging the momentum flux for that specific one for the duration of the event over the airglow images where it shows up, and then averaging over all the distinct wave events, just would be more laborious. We have a clustering algorithm that does work for us, but still, we choose not to do it here.

Regarding the SABER data, we did not acknowledge the SABER team in the proper section, but we should have. We have not offered coauthorship to the SABER team indeed. We are not aware it is required to offer coauthorship and thought an acknowledgment would suffice.
Fig. 1. Processing of a set of three airglow images in order to obtain the cross spectra. In (a) it is showed a set of sequential OH images presenting GW structures. (b) Time difference images obtained from the set of images. (c) Amplitude cross-spectra of the TD images, from where we estimate the wave amplitude, propagation direction and horizontal wavelength. (d) Phase cross-spectra of the TD images showed in (a).

Fig. 1.

---

## Author Comment (AC1) · 1 Dec 2020

We would like to thank all the referees for raising important comments about our manuscript. We have done our best to address all of them.

Please find attached the pdf where we provide a point-by-point reply to the comments.

Best regards

Please also note the supplement to this comment:

[Figure]

https://acp.copernicus.org/preprints/acp-2020-896/acp-2020-896-AC1-supplement.pdf

---

## Author Comment (AC2) · 3 Dec 2020

I have updated replies to two referee #1 comments. The ones presented in the "Full reply to referees" documents were not consistent with what was done in the paper.

line 293-294: The last sentence is enigmatic. If these larger amplitude waves are only seen in the O2 emission but not below (OH, Na) or above (OI), it is quite puzzling. • The fact that large amplitude waves are seeing in the O2 layer is surprising given the layers overlapping structures. This must be investigated separately. At this point, we

don't have a good explanation. However, the O2 has the narrower estimated FWHM for the campaign, and that would allow shorter vertical scale waves to be seen in the O2 images, and consequently larger momentum flux waves would be measured there (see lines 311 -313). We have added to the text the following statement:

"We have added the following to that sentence: "It is not clear why the enhanced waves are seen most in the O2 emission once the layer's peaks nearly overlap, but this could be related to the fact that the O2 VER has the smallest FWHM (see Table 2). These shorter $\lambda z$ waves would be seen in images of the O2 emission primarily, and their momentum flux would be larger for it increases as $\lambda z$ decreases (see Fig. 5i)"".

Figure 7: Can you add a detection threshold? • We have fixed that. We have added a horizontal line as a reference for the horizontal threshold.

[Figure]

**Fig. 1.**

The three plots show:

**(a) Weighted Horizontal Wind Fluctuation Spectra, O($^1$S)**

PSD ($m^2/s^2$) vs Frequency (cyc/hour), with curves labeled zonal and meridional, a 99% significance level line, and markers at 24h, 12h, 8h, 6h.

**(b) Weighted Horizontal Wind Fluctuation Spectra, O$_2$**

PSD ($m^2/s^2$) vs Frequency (cyc/hour), with curves labeled zonal and meridional, a 99% significance level line, and markers at 24h, 12h, 8h.

**(c) Weighted Horizontal Wind Fluctuation Spectra, OH**

PSD ($m^2/s^2$) vs Frequency (cyc/hour), with curves labeled zonal and meridional, a 99% significance level line, and markers at 24h, 12h, 8h.

---

## Author Response (AR1)

**Received and published: 5 October 2020**

This manuscript presents a fairly detailed investigation of gravity waves observed in airglow image data obtained during an observation campaign in November 2018, from Northern Germany. It is well-written and provides a good assessment of the effects of GW on the mesosphere, even if it is during a limited period of time. However, the authors should address the following would comments:

• We thank you for raising these important comments and suggestions. We address your comments the better we can.

- It is a little bit surprising that there are no waves with intrinsic period <20 min. Most of the waves should be propagating against the wind, therefore their intrinsic phase speed should be larger than the observed one, and their intrinsic period should be smaller than their observed one.

- In figure 5f, the reviewer can see that there are waves traveling against the wind, waves traveling nearly perpendicular to the wind, and in few cases, waves propagating to the same wind direction. Although there are a few waves traveling to the same wind direction because their larger phase speed, these waves do not transport significant momentum flux, as presented in figure 5a showing large momentum flux waves moving against the wind.
- We did not detect waves with periods <20 min, although they must be present certainly. Due to the filter wheel cycle time, a given airglow emission is sampled every 10 min, which allows only the detection of waves having periods >20 due to the long sampling period of 10 min each image is acquired.

- lines 255-271: It is complicated to compare the results with Li et al., 2011, because of the time resolution (~10 min) which does not allow for detecting most short-wavelength, short-period waves. As also noticed in the manuscript, this campaign covered only a few days, while Li et al. results represent the whole year. Same problems with Li, 2011.

• We did the comparison because the paper utilizes the same autodetection technique. we focus on the results of Li et al which are comparable to the wave scales we have detected. A stronger correspondence between the results is found in the longer horizontal wavelength range (100 kilometers or longer), which are associated to convective sources far away from the observation site. In this way, we can verify that our findings are in agreement with those of Li at al. (2011), even in the case we do not detect the shorter period waves, shorter horizontal scale waves.

- lines 284-285: These mean values are very small, especially considering the error. What does -0.36+/-1.51 mean? - 0.36+1.51=1.15, so the MF is carried in the other direction? Or you are sure about the wave direction of propagation, and so the minimum possible value is 0? I am not sure how it is possible to measure such small MF values giving the uncertainties in the measurements.

• The summary box in Fig. 5 shows the MF mean and MF standard deviation of the sample, not the error in the MF determination. Notes that the error in momentum flux must take into consideration the errors in all the wave parameters determined by our auto detection method for wave event (see Vargas, 2019). As we showed and discussed in the manuscript, most of the waves carry small momentum flux, and this fact reflects on the mean, which is very small. The standard deviation shows more or less the spread in the data, that is, the distribution of waves going to one side against those going to the opposite side.

- line 286: The total value is very misleading and cannot be used for any future comparisons. With the analysis method used in this paper, GWs are measured for each image, therefore the more images, the larger the total MF. If the imager cycle was 1 min instead of 10 min, the total MF would be 10x larger because 10x more waves parameters would have been measured! An average MF value for each wave and its duration could be more useful to assess its impact. The statistical results (11% of the waves responsible for carrying 50% of the MF...) are still relevant but the authors have to be very careful when giving the "total" MF. -

• I agree the total MF value means nothing and will cause confusion. To give a meaningful estimation of the total momentum flux, we should have found the number of independent waves (see our reply to your next comment). We have removed the total MF from the paper.

The number of waves detected (362) is also controversial. There were NOT 362 separate GWs propagating over the observation site during these 4 nights! There were 362 wave measurements, which is different. Some of the waves probably lasted for hours and their parameters were measured several times. Of course, these parameters can evolve depending on the forcing and the background atmosphere conditions, but there were still the same GWs coming from the same sources

- The waves are not necessarily independent. We have defined as a gravity wave event the output of the cross-spectrum of two time-difference images when the special peak is larger than 10% of the total energy of the spectrum. two time-difference images are generated from three light frames of the airglow. Thus, if a wave is detected in a given set of three light images, it is considered an independent wave detection. If, in the next set, another wave detection is made, the only way to tell the two events correspond to the same gravity wave is by comparing their wave parameters, which vary among image sets. Now, the momentum flux of a wave varies as it goes through the field of view as well. Thus, considering the mean MF of all the waves detected during the campaign would be the same if we cluster the corresponding wave events into a distinct wave, average the momentum flux for that specific wave for the duration of the event over the airglow images where it shows up, and then average over all the distinct wave events found. This less efficient and more laborious though. We have a clustering algorithm that does that work for us, but still, we choose not to do it in the present manuscript.
- As we rely on a set of three images at the time from where we obtain two time-difference images, and finally the cross spectra from the 2DFFT of the time difference images. So, the time span of each set is about 30 min, and most of waves would have completely disappeared the imager FOV within that time span because the duration of a gravity wave is short based on our experience. This way, it would be more likely that each detection corresponds to an independent wave detection.

The GWs were observed in 4 different airglow layers, at 4 different altitudes. Could you give more information about that? Which waves were observed simultaneously? It would also be interesting to see the difference in MF between the airglow layers.

• The momentum flux of waves in different layers can be given and we have estimated it separately, but because the detections in an individual layer were not too many, we showed the results together. We have provided separated plots analogous to figure 5 as supplementary material where the reviewer can see the wave detections over time in individual layers.

The MF divergence would be very interesting, as well. - You should mention ducted waves. Even if the waves presented in this paper are unlikely to be ducted because of the minimum intrinsic periods, these waves exist but do not carry any vertical MF, so they would bias your results. Again, a comparison between the different airglow layers would help

• Estimating the momentum flux divergence between two layers is possible and we have done that in Vargas et al, 2015, http://dx.doi.org/10.1016/j.asr.2015.07.040. The simultaneous identification of a wave in two layers is more demanding, and we do that for specific, individual cases. Instead of that, here we have estimated the flux divergence from our model (Vargas et al, 2007, JOURNAL OF GEOPHYSICAL RESEARCH, VOL. 112, D14102, doi:10.1029/2006JD007642, 2007) based on the distribution of periods and wavelengths of the observed waves. In this paper, we have done that as if the wave amplitude was measured in the OH and O(1S) layers simultaneously, with no change in amplitude detected (saturated waves).

- It is probably beyond the scope of this paper, but given the small number of nights, looking for sources (tropospheric weather, convection, fronts...) would be interesting.

• We did not address the sources of the waves in this paper but will do it in a follow up publication exploring the sources of primary and secondary waves based on the possibility of secondary wave generation in the range of 30-40 km as suggested by lidar temperature data.

Minor comments:

line 12: can you add the number of nights/hours of observation? It would give a better idea of the importance of the MF values.

• The number of hours per night can be seen an estimated from figure 1 keograms. We have estimated that and added to the text in line 10 (four nights, 45 hours nighttime observations).

line 26: remove "successfully"

• We have removed that.

line 93: long-period variations Section 2.2: what are the vertical and temporal resolutions of the radar system?

• The MSMR data was processed with vertical resolution of 1 km and temporal resolution of 30 min for the total wind field (Fig2a). The fluctuations (Fig2b) were calculated by subtracting the total wind field from a heavily averaged wind with 4-km vertical resolution and 4-hours temporal resolution over 400 km FOV of the MSMR system.

line 137: FWHM

• We have corrected that.

line 175: "duration" instead of "length"

• We have corrected that.

line 177: The MF equation is proportional to Lz/Lx, so it is not that surprising that large horizontal wavelength waves carry less MF. It is more surprising that large vertical wavelength waves carry less MF, though.

• From Vargas et al. (2007), we see that the momentum flux (Eq. 15) is given by  $F_M/\rho_u = -\frac{1}{2} \frac{\omega^2 g^2 m}{N^4 k} (\frac{l'/l_0}{CF_I})^2$ , and other variables like the cancelation factor (vertical wavelength dependent) and the wave amplitude also are taken into account. Observe that FM has a square dependency on these variables as well as the wave frequency. All in all, because MF is multidimensional in the wave parameter variables and CF, there is no easy way to visualize it in a plot. Maybe would be optimum to show layers where two FM parameters vary while the others are kept constant.

line 191: You must have done your filtering wrong if the cutoff was 5 hours and you still see that many peaks, especially the ones around 6, 8, and 12 hours! The peak at 8.9 hr is pretty close to the peak at 8 hr. You have to give the resolution of the spectra, otherwise this is not very convincing.

- We have not explained correctly the way the wind fluctuations were calculated (it is corrected now in the manuscript). To obtain the wave fluctuations (Fig2b), we first calculate 4-hours, 4-km temporal and spatial windows to average the wind over 400km field of view. then we subtract the result from the total wind field. The total wind field was calculated using much smaller temporal and spatial windows, that is, 30 min and 1 km, respectively. Because that, the resulting fluctuations still keep some of the energy of larger periods waves as can be seen in the spectrum.
- Although we are reducing the effects of tides and waves with periods larger than 4 hours using: (a) temporal filter of 4 hours, (b) altitude filter of 4 kms, and (c) the inherent horizontal filter of about 400 km or so, we can see that signature of 8 hours with large vertical wavelengths and very large horizontal wavelengths are still visible.
- Looking to the spectrum alone we cannot tell apart tides to gravity waves. However, we can identify larger periods gravity waves in the airglow because their horizontal wavelength is ~1000 km whereas the tidal components have much larger horizontal scales. This way, the ~1000 km, 8 hours waves can be seen in the airglow keograms as the tilted brightness structures as those in Fig. 8b.

line 204: The keograms could be improved by flat fielding the images first. Not sure if this had been done, but it would improve the signal at the zenith compared to the edges. Figure 8a bottom is misleading. It looks like the

wave tilt (which is related to the direction of propagation and phase speed) corresponds to the decrease in meridional wind intensity. Not sure how to avoid that. You should add the directions of propagation in Table 3.

• We perform all the pre steps, flat fielding included, before calculating the 2DFFT of the keograms. By simulating a 3D gravity wave perturbing the airglow and using the gravity wave polarization relations from the linear theory, we would be able to confirm that the airglow intensity increases with the decreasing of the meridional wind.

line 221: What is the error on the horizontal wavelength measurements?

• The uncertainty in the horizontal direction can be determined from the horizontal wavenumber spectrum where the spread of the wavenumber energy that leaks to adjacent wavenumbers. From our calculations in Vargas 2019, we have estimated more or less 10% uncertainty (included in the text now) in the horizontal wavelength, that is, ~130 km for the wave seen on Nov. 3-4.

line 235: 8 hours, could be a tidal component.

• As pointed out above, the large-scale wave of Nov 6-7 does not correspond to a tidal component since the horizontal wave structure can be fully seen in the keogram of Fig.8b.

line 293-294: The last sentence is enigmatic. If these larger amplitude waves are only seen in the O2 emission but not below (OH, Na) or above (OI), it is quite puzzling.

- The fact that large amplitude waves are seeing in the O2 layer is surprising given the layers overlapping structures. This must be investigated separately. At this point, we don't have a good explanation. However, the O2 has the narrower estimated FWHM for the campaign, and that would allow shorter vertical scale waves to be seen in the O2 images, and consequently larger momentum flux waves would be measured there (see lines 311 -313).
- We have added the following to that sentence: "It is not clear why the enhanced waves are seen most in the O2 emission once the layer's peaks nearly overlap, but this could be related to the fact that the O2 VER has the smallest FWHM (see Table 2). These shorter λz waves would be seen in images of the O2 emission primarily, and their momentum flux would be larger for it increases as λz decreases (see Fig. 5i)"..

Lines 363-365: Can you rephrase this sentence? It is not very clear.

• We have removed the referred sentence because it was too speculative.

Line 378: The links are missing!

We have fixed that.

Figure 4: (d) Calculated volume emission rates...

We have fixed that.

Figure 7: Can you add a detection threshold?

• We have fixed that. We have added a horizontal line as a reference for the horizontal threshold.

The comparisons with previous publications are lacking some relevant studies: eg, Bossert et al., 2015, for MF of long period waves, Cao and Liu, 2016, for the impact of large vs small MF GWs, and even on the same topic Hertzog at al., 2012, Plougoven et al., 2013, or Wright et al., 2013. The last 3 describing stratospheric measurements, though. The comparisons with Li, 2011, and Li et al., 2011, are not very relevant except because they used a similar analysis method.

- We explained earlier and offered other reasons of why we did compare our results with those of Li et al. (2011) and Li, 2011.
- We have looked into the suggested references and have decided to use that of Cao and Liu, 2015 and Bossert et al. (2015) to compare with our results. We appreciate the suggestions.
This manuscript deals with gravity waves detected using an airglow imager and dis- cusses their effect on the background wind, combining collocating meteor radar net- work data and satellite data as background information. Their effort to make full use of the obtained data during the limited campaign period is much appreciated: expanding their analyses to waves with horizontal scales larger than the field of view of the imager. However, the overall analysis methods often lack detailed explanation and hard to follow, and some part of them even seem inadequately done. The authors are advised to revise the manuscript substantially by addressing the following comments.

- We cannot thank enough the reviewer for taking his/her time to read our manuscript and bring up important comments and suggestions to improve it.
- We recognize we need to expand the description of the data analysis to make it clear for the readers. Also, we believe the keogram spectral analysis would be understandable as we wrote, but it is obvious we need to specify it in more detail in the manuscript.

**Major comments:**

The sequence interval of the imager observation is 10 min, and the exposure time is mostly 2 min. This means that possible aliasing should be taken into account in the analyses although it is not described in the manuscript. Because of the 2 min exposure, any structure with a period shorter than 4 min will be smoothed out and not be resolved, but those with a period between 4-20 min can easily alias into slower period motions, and significantly affect the results.

• We believe that, because every image of a given airglow layer is taken at 10 minutes pace (the imager filter wheel cycle), we are only able to resolve waves with 20 minutes or longer. However, we believe the aliasing is minimal in this case because the small-scale wave structures (horizontal wavelength less than 725 km) seen in the airglow images are sharp (no smudging).

In detecting wave structures from wind data, a high-pass filter with a cut-off of 5 hr is used. However, the detected wave periods are around 4 and 8 hr, which are close to or longer than the cut-off. The wave amplitudes and also phase structures will be largely affected. Even a spectral analysis is done using the filtered wind, and fluctuation components with periods longer than 4 hr are discussed (Fig 7), leading to the subsequent analyses of those components based on the filtered wind as shown in Figures 8 and 10. Although the authors mention these peaks as 'leakage' (line 190), I am afraid that such data treatment is not acceptable at all. What the authors can remove before the spectral analysis in the present case would be only the background wind (mean wind) and trend. Since the imager data is unfiltered, wind values should be treated in a similar manner in comparison.

The details of 2D spectral analysis method shown in Fig 9 are not clear. The keograms shown in Figure 8 are thought to be used, of course, but how is the calculation done? The borders of the keograms can significantly affect the results if miss handled. Isn't there a possibility that the evening twilight seen around 5 pm affects the results? In other words, what kind of spectral window is applied in the analysis?

- We should not have referred to that as "filtering" because reader will think it as a digital filter operation. Here is how we did it: The total wind field was obtained by processing the raw meteor radar data using a 30-min, 1 km window. To calculate the wind fluctuations, the raw meteor radar data was processed using 4-hours, 4-km averaging windows. Then result was than subtracted from the total wind field resulting in the wind fluctuations associated with gravity waves. We have corrected this in the paper to make it clear for the reader what has been actually done to obtain the wind fluctuations.
- We believe the spectral analysis of the wind fluctuations is not faulty. We did not explain correctly how the wind fluctuations were calculated, but this issue is fixed now. Figure 7 shows indeed the relevant wave amplitudes during the campaign, even though we could not suppress completely the tidal modes. A

significance threshold line (99% confidence) has been added to the plot in figure 7 to show the relevant wave amplitudes and periods.

• We have addressed the issue with the filtering of the horizontal wind for the wind fluctuations. The filtered wind is used to compare with the large-scale waves in the airglow keograms. For correcting the wave period Doppler Shift, we use the total wind at the moment of wave detection, not the filtered wind nor the fluctuations.

Minor comments: Abstract

Line 1 'large and small'

• We have changed that.

Not quantitative. More specific description is wanted. The same comments apply to the other 'small' and 'large' in the manuscript.

- We have provided a quantitative estimation of what large and small horizontal scales mean in this paper (line 130-135).
- The small-scale events refer to waves with horizontal wavelength <725 km, while large-scale events are considered here as having horizontal wavelength >725 km. The 725 km range corresponds to the diagonal line across the field of view of an image mapped into a 512x512 km2 grid. This way, a 725 km wave would still be fully seen within the image frame (on crest, one trough).

**Line 6 'flux divergence estimations'**

Momentum flux is estimated, but the divergence does not seem estimated, instead MF is just assumed to be deposited around the mesopause.

• We have calculated the momentum flux of each wave detected in airglow during the four nights of the campaign because the auto-detection method outputs all the wave parameters necessary for the MF calculation. Thus, the momentum flux was calculated for each wave individually. However, the flux divergence was estimated from our simulations carried out in Vargas at al. 2007. The divergence estimations assume the waves are saturated, meaning that the wave amplitude doesn't change between two airglow layers. The assumption that the majority of waves are saturated is based on lidar measurements perform over Maui and SOR Observatory in New Mexico. In our reply to reviewer#1, we have explained how we would calculate the flux divergence directly from the airglow images. We would have to follow an individual wave to verify if it's visible in two layers simultaneously, then we would measure the MF of this wave in each layer, and finally subtract these values to have an estimation of the divergence for that specific wave. However, we have not done that in this paper.

**Line 11 Mean MF, total MF and number of detected events**

585.96/362 is not 0.88.

- The reviewer is correct. We have updated the estimation to 1.62+-2.70 m^2/s^2.
- We have realized that showing the total momentum flux of all waves detected brings more confusion (see reply to reviewer#1) and decided to remove that from the manuscript.

How is the mean estimated? Is the detected number 362 the number of independent wave events, or multi-counted number? The same comment to the lines 285-286.

- The waves are not necessarily independent, but it would be more likely that each detection corresponds to an independent wave detection.
- As we rely on a set of three images at the time from where we obtain two time-difference images, and finally the cross spectra from the 2DFFT of the time difference images. So, the time span of each set is about 30 min, and most of waves would have completely disappeared the imager FOV within that time span because the duration of a gravity wave is short based on our experience.

Lines 17-18

The lidar data in the upper stratosphere is not actually analyzed in the manuscript, but only qualitatively mentioned. This statement does not seem appropriate in the abstract.

• We have removed that.

The main body of the manuscript

Line 52 'modes'

Meaning not clear.

• We have reworded the sentence for clarity.

Line 58 'large amplitude filtered wind fluctuations' Meaning not clear.

• We have removed the word filtered to avoid confusion.

Line 84 'long period oscillations'

The wave structure is not very clear. Even one cycle is hard to distinguish, especially for the Nov. 06-07 case.

• The Nov 6-7 case is less pronounced, but can be seen in the zonal keogram around 1200 UTC. We have overlayed ellipses to make it easier for the reader to identify the airglow structures. It is clear that we have only considered these structures because they have also caused pronounced fluctuations on the wind fields and are associated with those large-scale waves, otherwise it would be hard to prove and explain that they would be gravity waves indeed.

Line 91 'time difference'

Is this done using pairs of the adjacent 10 min separated images? The detail should be written.

• We have put details of the image processing in the appendix, one for the auto-detection method, and other for the keogram analysis. We hope this makes this part clear for the reader.

Line 116 'smaller' > shorter

• We have rewritten the sentence. Line 116 'simultaneously the back....'

'with' after 'simultaneously'

• We have rewritten the sentence.

Line 118 'Fig. 1c'

Meant to be Fig. 1d?

• It should be meridional keogram in Fig. 1c. The figure 1d refers to the small-scale waves (lh<725 km). Section 3

The definition of 'small' and 'large' should be described at the beginning of Section 3. Sub-section 3.2

• We have done that as asked. The small-scale events refer to waves with horizontal wavelength <725 km, while large-scale events are considered here as having horizontal wavelength >725 km. The 725 km range corresponds to the diagonal line across the field of view of an image mapped into a 512x512 km^2 grid. This way, a 725 km wave would still be fully seen within the image frame (on crest, one trough).

The meaning of 'fluctuation' at line 182 is not clear although it is inferred later at lines 190-191. It is confusing. The filter characteristics also should be written in the caption of Figure 6.

We have removed the word 'fluctuation' for simplicity since the result of the weighted wind calculation can be seen in the figure. The details of how the wind fluctuations were obtained are also added to the caption of Fig. 6.

Lines 210, 211, 219, 220, 224, 234, 235 316, 317, 320 and 321

Why are the uncertainly of all these values '+-1'?

• The uncertainty corresponds to the smallest division in the keogram temporal axis. Line 244 'amplitude'

Since the background wind is not thought to be a wave, rewording will be appropriate; something as magnitude.

• We agree with the reviewer and have changed that. Line 245

Absorption occurs at a critical level, but reflection occurs at a turning level. Different wind structures are responsible, perhaps as you know.

• There is confusion in the literature. I think as critical level the altitude where the wave could be absorbed or reflected, as the word 'critical' suits both. If we would specify further, then we should use turning level to the altitude where the wave is reflected, and absorption level the altitude where the wave is absorbed. It is much clear and gives no room for confusion. We have removed "critical" and use only absorption or turning lever to be precise of the meaning we want to convey.

Lines 283-284 'toward the west of -0.36+-1.51 m2/s2' Very confusing description.

• We have detailed further the explanation.

Lines 286-288 The finding of a small number of waves carrying a large portion of MF is a good point. I suggest that the authors include discussion on GW intermittency and its importance in model atmosphere studies.

• We have added the discussion about wave intermittency.

**Lines 289-294**

This can largely affect the obtained results. Hopefully more discussion wanted. Isn't there any possibility that the layer thickness is related, or is it already taken into account in the analysis? It seems from Figure 4 that O2 layer is the thinnest.

Wave detections in separate layers do not rely on the information about layer thickness or layer altitudes but only in the amplitude of the wave that must be large enough to produce a prominent peak in the cross spectrogram of amplitude. Thus, those features (layer thickness and altitude) hardly would influence the results. However, the fact that large amplitude waves are seeing in the O2 layer is surprising given the layer overlapping structures. This must be investigated separately. At this point we don't have a good explanation and providing one right now would be pure speculation. However, the O2 has the narrower estimated FWHM for the campaign, and that would allow shorter vertical scale waves to be seen in the O2 images, and consequently larger momentum flux waves would be measured there (see lines 311 -313).

**Line 296**

Yes, if the acceleration or deceleration occurs all along the same latitude circle, to my understanding.

• We could not fully understand the reviewer point of view here. We could provide a better answer if the reviewer is willing to elaborate further this comment.

**Lines 339-340**

I think this is a nice point. In the radar studies, a long-lasting topic has been which period range of GW is most responsible for MF deposition. Short period GWs were thought to be responsible before, but recent studies suggest that longer period waves are responsible. One example I know is Sato et al, 2017

(https://doi.org/10.1002/2016JD025834). There can be more. Including such preceding studies will be beneficial for even more fruitful discussion.

• We have added a discussion as suggested and also added the Sato et al. (2017) paper to the reference list. Figure 2 '> 5 hours' should be '

**General comment:**

The authors describe different measurements they carried out during the SIMONe-campaign in November 2018 at or around Kühlungsborn, Germany. Supplementing satellite-based data are used. They combine the measurements in order to fully characterize gravity waves and discriminate between large and small-scale waves. However, they do not explain what they understand under small and large scale.

From my point of view, there are two main results. The authors found out that approximately 11% or all detected waves carry ca. 50% of the total momentum flux and they had a specific look on two large wave events. They motivate that these structures might be due to secondary gravity waves. Nevertheless, this remains speculative. The manuscript is well-structured and sections 1 to 4 are easy to read in the first approach. However, on closer inspection important information is missing. In section 5, the formulas disturbed my flow of reading. Following the classical approach, they should appear earlier in the manuscript which would help here. The data and the methods used to measure and identify the gravity are appropriate, however especially the data section could have some more details. My main point concerning the section results refers to error bars and/or significance levels: I didn't find a single one.

I propose to accept the manuscript after some major revision.

• We thank the reviewer for raising important comments about our paper and for taking the time to do so. We appreciate your help very much.

**Major points**

The manuscript is written in a rather abbreviated style.

The most important point in this context is that the manuscript suffers from a lack of information about the quality of data and analyses.

• We understand the reviewer's concern about having written the manuscript in an abbreviate style. Maybe because we wanted to make it concise, we have failed while describing important parts of the analysis and processing, which must be done precisely to permit the reproduction of the results.

I found neither a single error bar nor a significance test in the section results (in the discussion part there are very few error bars). For clarity issues, it is probably not helpful to provide this information directly in all figures (for example I have no idea how to incorporate error bars in keograms), but it can be done in some cases (FFT) and in the others the information can be given in the text. Parts of the main results depend on statistical analyses.

• The incorporation of error bars in the spectral analysis could be found from our simulations of the error propagation into gravity wave parameters in our recent paper (Uncertainties in gravity wave parameters, momentum fluxes, and flux divergences estimated from multi-layer measurements of mesospheric nightglow layers, https://doi.org/10.1016/j.asr.2018.09.039.)

A second point in the context here is that the authors do not explain how they define wave event and which consequences this definition has on their results. If I operate an imaging system which takes a picture every ten minutes, is each result of the spectral analysis of each image defined as a wave event or is a wave event an oscillation in space which shows nearly the same horizontal wavelengths over some time (so in some images). Fast waves would be underrepresented in the first case. Does this have any influence on the results?

• For the small-scale analysis using the autodetection method, we have defined as a gravity wave event the output of the cross-spectrum of two time-difference images when the special peak is larger than 10% of the total energy of the spectrum. Two time-difference images are generated from three light frames of the

airglow (please take a look at fig 01 for Vargas et al 2009 (https://angeo.copernicus.org/articles/27/2361/2009/) showed here below:

Figure 1 Processing of a set of three airglow images in order to obtain the cross spectra. In (a) it is showed a set of sequential OH images presenting GW structures. (b) Time difference images obtained from the set of images. (c) Amplitude cross-spectra of the TD images, from where we estimate the wave amplitude, propagation direction and horizontal wavelength. (d) Phase cross-spectra of the TD images showed in (a).

- Thus, if a wave is detected in a given set, it is considered an independent wave detection. Now, if in the next set another wave detection is made, the only way to tell the two events to correspond to the same gravity wave is by comparing their wave parameters. Now, the momentum flux of a wave varies as it goes through the field of view. thus, considering the momentum flux average of all the waves detected during the campaign would be the same as clustering the corresponding wave events into a distinct wave, and then averaging the momentum flux for that specific one for the duration of the event over the airglow images where it shows up, and then averaging over all the distinct wave events, just would be more laborious. We have a clustering algorithm that does work for us, but still, we choose not to do it here.
- Again, we rely on a set of three images at the time from where we obtain two time-difference images, and finally the cross spectra from the 2DFFT of the time difference images. So, the time span of each set is about 30 min, and most of waves would have completely disappeared the imager FOV within that time span because the duration of a gravity wave is short based on our experience. This way, it would be more likely that each detection corresponds to an independent wave detection.

Another point is: It is really strange to see a paper where TIMED-SABER data are used and Martin Mlynczack and James Russell (or at least the SABER team in general) are neither co-author nor mentioned in the acknowledgements. There isn't even a single citation of them (e.g. Russell et al., 1999,

https://doi.org/10.1117/12.366382 or Mlynczak, 1997, https://doi.org/10.1016/S0273- 1177(97)00769-2). On the other hand, there exist co-authors here who also "only" delivered data. This is not consistent. Maybe the authors offered co-authorship to the SABER people and they declined, I don't know that, but is there really nothing to mention in the literature list or in the acknowledgements?

• Regarding the SABER data, we did not acknowledge the SABER team in the proper section, but we should have. We clearly thank the SABER team now in the acknowledgements statement. We have not offered coauthorship to the SABER team indeed and are not aware coauthorship is required to use SABER data. We

think an acknowledgment note will suffice. We have also included the suggested references to the list and made citations at proper locations.

**Minor points**

**Section 1**

General comment: The introduction is quite general. In my opinion, the technical focus of this manuscript is on the combination of different measurements to describe gravity waves as comprehensively as possible. Airglow imagers are used in order to derive horizontal wavelengths and periods. Wind information come from radar data. Temperature information are based on SABER data. Other groups have already done such studies or similar ones - just enter 'airglow radar SABER gravity waves' in google scholar. However, there is no reference in your manuscript. One of the main topics of the manuscript is the derivation of the momentum flux. There are already other publications on this topic which are not mentioned here or in the discussion (e.g., Ern et al., 2011, https://doi.org/10.1029/2011JD015821 derived the momentum flux globally based on satellite data). Please include some citations and put your work into the context.

• We have included citation of other references including momentum flux derivation in the introduction section, including that on of Ern et al, 2011 and others we could find.

p. 2 l. 48 Can you please provide the exact campaigns period?

• We have done so.

p.3 l. 55 & 57 What is small and large scale for you? This comments also refers to the headings of section 3.1 and 3.2.

The small-scale events refer to waves with horizontal wavelength <725 km, while large-scale events are
considered here as having horizontal wavelength >725 km. The 725 km range corresponds to the diagonal
across the field of view of an image mapped into a 512x512 km2 geographical grid. This way, a 725 km wave
would still be fully seen within the image frame (on crest, one trough). We have also included these
specifications in the text.

**Section 2**

From my point of view, the level of detail of the different subsection varies (2.2 has more details than the two other subsections). Additionally, I do not get all "classical" information for all instruments such as spatial and temporal resolution or some information about the data quality (not necessarily bias and precision but for example comparisons with other instruments, for SABER such studies are available).

I also miss further literature for ASI and SABER where the reader can look up (technical and data processing) details about the instruments (e.g. concerning ASI the detector size etc., concerning SABER the retrieval etc.). For SABER, different data versions exist. I conclude from the legend of figure 4 that you use the newest one (2.07) but not every reader might be familiar with the "SABER notation", so please provide this information also in section 2.3.

• We have done our best to attend your request.

p. 4 l. 87 In the text you use UTC, in figure 1 you use p.m. If it was consistent it would be easier to read.

• We understand your concern, but, respectfully, we have decided to keep it as is because this minor issue will not affect the interpretation of the plot and the results of the paper.

p. 4 I. 94 You write that the result of a time-difference operation is an image where the contrast of small-period, small-scale oscillations is enhanced. I think with small-scale you mean the spatial dimension. In this case, I do not fully agree with you. Whether you enhance a signature or not should depends on the speed of the signature: if it is near zero you won't see it in your time-difference image because you just subtract it. The phase speed is directly proportional the wavelength (in this case the horizontal one) and indirectly proportional to the wave period. The phase speed significantly differs from zero, the larger the wavelength or / and the smaller the period is. So, you enhance small period but large-scale oscillations.

• The reviewer has presented a good point. Our definition of small horizontal wavelength (lh) wave considers lh

*Figure 2* Ratio between wave amplitudes observed in a TD image and in an original image as a function of the intrinsic period of the wave. *The sampling period used to compose this graph was 2 minutes.*

p. 4 I. 115 Which high-pass filter exactly? If you say you use a five-hour high pass filter, it means for my that all periods longer than five hours can pass. Later in the document, it becomes clear that the opposite is true and you probably referred with the term "high-pass" to the frequency domain. Can you please clarify here which periods are still present in the filtered data. Additionally, in the spectra in figure 7 you still see a rather dominant p. 4 I. 118 Are you sure that you really see exactly this oscillation?

- We should not have referred to that operation as filtering because reader will think as a digital filter operation. Here is how we did it: The total wind field was obtained by processing the raw meteor radar data using a 30-min, 1 km window. To calculate the wind fluctuations, the raw meteor radar data was processed using 4-hours, 4-km averaging windows. Then result was than subtracted from the total wind field resulting in the wind fluctuations associated with gravity waves. We have corrected this in the paper to make it clear for the reader what has been actually done to obtain the wind fluctuations.
- From this, it is clear that oscillations of periods longer than 4 hours will be suppressed, but not completely
  eliminated. Also, waves having vertical scales <4 km will be suppressed, but not completely eliminated in the
  resulting wind fluctuations. The spectra in figure 7 reflects the fact that not all longer period oscillations were
  eliminated, but it doesn't mean the spectra is incorrect. We also can tell what peaks are associated with gravity
  waves once waves of similar periods are also seen in the keograms.</li>
- p. 5 l. 120 The abbreviation SABER is not explained in the manuscript.
- We have explained the abbreviation earlier in the manuscript when we first mention the acronym SABER in line 59-60.
- p.5 l. 127 Please concretize the lapse rate you mean.
- The reviewer has asked to explicitly give the value of the lapse rate, is that correct? See lines (131-133.
- p. 5 l. 127 Please be a little bit more precise here. It is not important that the orbit of TIMED is exactly over the observatory, the field of view of the instrument has to measure over or near the observatory.
- We have elaborated more. The specific fragment now reads "Even though the satellite orbits registered during SIMONe-2018 were not exactly over the observatory, the instrument measurements are performed in limb vertical plan that is near or within the field of view of the imager (Fig. \ref{fig03}), where the colored dots indicate where the measurements are being made, not the satellite position"
- p. 5 l. 128 From figure 3 it looks like you use also profiles which were not within the field of view of the imager.
- The reviewer is correct. We have used all the measurements we could find to diagnose the background atmosphere stat in the nighttime period during the campaign.

p. 5 l. 133 It is great that you are able to calculate the VER profiles for the different species but why don't you use the ones provided by SABER?

- The reviewer is correct. That is because we did not want to mix in the analysis the use of measured profiles with the calculated profiles. It would not cause a big change in the results; thus, we carried out the analysis using only the calculated profiles.
- p.5 I 137 Please correct the letter shift (FWMH→FWHM) and explain the abbreviation.
- We have done so.

**Section 3.1**

The way to present such campaign results is always a tightrope walk. If the reader is flooded with details, he loses the overview, if it is too short, questions remain open. For me questions remain open.

I miss two formulas (dispersion relation for high-frequency waves and vertical flux of horizontal momentum, or at least citations of literature where the reader can look them up) and in the case of the dispersion relation I would like to read some information about the Brunt-Väsälä frequency (probably calculated based on SABER, VER weighted?).

• We have addressed the lack of explanation by providing two appendix sections where we detail the autodetection method and the keogram spectral analysis for the readers.

Please also provide the information that the cross-spectrum is the Fourier transform of the cross- covariance function, since you treat pictures you probably use the 2D version. Another useful information would be the sensitivity of the analysis. You calculate the cross-covariance of two TD images. The calculation of TD images already means filtering the data for fast waves. The calculation of the cross-covariance provides you information about the waves which change from one TD image to the other. So, there are two stages of filtering and it would be interesting to provide information which waves pass this filter, and which don't.

- The reviewer is correct, we have indeed used do 2D Fourier transform in each time difference image and then combine the spectrum of these two obtaining the cross spectrum the figure 1 above shows the process clearly.
- One thing that is not clear in the figure is that we have correct the images (shifting pixels) to account for the wind velocity across the field of view, that is, we have performed the Doppler shift correction in the wave period already. This process is depicted in the figure 3 bellow. Notice the red arrows indicating the shift in a bright spot of the image.

---

## Author Response (AR2)

Report #1
Submitted on 25 Jan 2021
Anonymous Referee #1

Thank you for addressing my suggestions to improve your paper.

We deeply appreciate your comments which have helped to improve our paper enormously.

I have a few more comments about your answers and the edited text, though.
- I will try to better explain my previous comment about the 20-min intrinsic period. If a wave has an observed period of 20 minutes (the limit due to the camera sequence), with an observed phase speed of for example 75 m/s (Lh would then be 90 km), but if it's traveling AGAINST the wind (25 m/s wind amplitude for example), the intrinsic period of the wave will be only 15 minutes. Since a lot of the waves you observed are propagating against the wind (Figures 5f and 5k), it is surprising that almost no waves have an intrinsic period < 20 min (Figure 5e).

I believe the answer for could be in the auto-detection method processing sequence. The Doppler shift caused by winds is corrected prior to obtain the wave parameters. Then we calculate the intrinsic wave parameters. I believe the scenario described by the reviewer would be true if the Doppler shift correction was done after the observed wave parameters were obtained. Ultimately, I believe it comes down to the sampling period of 10 minutes limiting the detection of periods longer than 20 minutes, which is a consequence of the Nyquist theorem.

- The time span of each set you analyzed is 20 min, not 30 (first image at time = 0, second at time = 10, third at time = 20).

- We agree with the reviewer and have corrected that in the paper.

- Furthermore, I don't agree that waves last only a short time and that they would have disappeared from the camera fov (512 km in your case) in 20 or 30 min (if the horizontal phase speed is 50 m/s, it would take the wave ~3 hrs to propagate 512 km!) Some GWs can be observed during the whole night as they depend on the source activity and the background atmosphere conditions. There are multiple publications about wave events lasting for hours, so you have to be careful when you consider that your wave detections are independent.

- The waves described by the reviewer resemble trapped oscillations, meaning these waves do not propagate vertically. The auto-detection method only outputs parameters of vertically propagating waves. Thus, the method samples waves that would move across the layer and disappear. For instance, taking our wave parameter distribution, the typical vertical wavelength is 20 km and typical period is 30 minutes, giving a vertical

phase velocity of 0.7 km/minute. It would take about 15 minutes for the wave to cross the 10 km width of the layer. That justifies our claim that the waves have short duration. This short duration events hit at either a short lifetime excitation source, or intermittent transmission source, which the autodetection method does not tells apart since the gravity wave parameters change from set to set.

- a more "manual" method would have been interesting to identify the waves as the data set is not that big (only 4 nights with images every 10 min).
  - The benefit of the autodetection method is that the human bias is removed from the wave analysis, meaning that manual analysis would bias the statistics toward large amplitude waves because they are more appealing to the eye. Observe that this human bias exists indeed as parametrization of gravity waves rely on statistics of large amplitude wave events, which are the minority of waves observed. The impact of small amplitude gravity waves in the MLT dynamics should be also taken into account.

- difference imaging can create bias as it tends to filter out waves which don't propagate (like mountain waves),
  - That is correct. Terrain-trapped oscillations and any non-phase propagating perturbations (like ripples) are attenuated by the time difference filter.

or waves with an observed period which is proportional to the observation sequence (in your case 20, 30, 40... min).
  - That seems not the case since our results show multiple waves in the 40 minutes range.

Minor comments:
line 7: characterized: corrected.
line 8: percentage: corrected.
line 21: transports (maybe, not sure, does it apply to "class" or "waves"?): corrected.
line 24: When these waves... : corrected.
line 51: ... to study large-scale... : corrected.
line 58: provided: corrected.
line 92: discusses: corrected.
line 107: you explain where 725 km comes from later, but maybe you should do that here: removed.
line 114: to receive: corrected.
line 133: present: corrected.
line 135: window. : corrected.
line 145: word missing after "each": corrected.
line 161: simulated: corrected.
line 162: ...oxygen profiles... : corrected.
line 181: 20 minutes: corrected.
line 183: ...it is more... : corrected.
line 192: during the campaign: corrected.
line 194: returns: corrected.

line 201: ... layers as the layer peaks are within... : corrected.
line 204: larger scales: corrected.
line 224 and Figure 8: can you change the time in the figure from am/pm to zulu (21, 22...)? : corrected.Same thing for Figures 1 and 10. : corrected.
line 247: reveals: corrected.
line 252: remove the extra "the": corrected.
line 254: on Nov. 6-7... : corrected.
line 262: Within the 86-98 km range... : corrected.
line 284: during a week: corrected.
line 289: which only allows: corrected.
line 291: ... minutes, near... : corrected.
line 299: for the results of... : corrected.
line 314: not sure they are so short. Do you have any references to back that up? No, this is based on our own experience.
line 315: As I wrote before, I think they are unlikely to be always independent. Please see reply above.
line 338: "presented" instead of "demonstrate": corrected.
line 341: "calculated" instead of "demonstrated": corrected.
line 349: revealed: corrected.
line 381: have also... : corrected.
line 415: second $L_z$ should be $L_h$: corrected.
line 415: ...will be smaller because... : corrected.
line 416: unlikely: corrected.
Maybe Appendix A could be included in the text. We have decided to keep it in the appendix and closer to the other technical s appendix sections.

As another reviewer pointed out concerning SABER data, it is surprising that nobody from Boston University is included as a co-author or at least acknowledged since they provided the image data.
Boston University does not own the imager and did not provided the image data, but only host the images generated by the system. We have added to the Acknowledgements: "We thank the support of Boston University colleagues and in particular J. Baumgardner to operate the airglow imager."
-

Report #2
Submitted on 25 Jan 2021
Anonymous Referee #3

I appreciate the efforts of the authors to answer my comments. From my point of view the quality of the manuscript improved a lot.

We thank you and appreciate the time taken to read and point out issues in our manuscript.

However, I am still a bit unsure about the analysis the authors applied for the derivation of the short-scale waves in the following sense:

• Thanks for providing a figure in the comments which shows the ratio of the amplitude measured in TD pictures and the original one versus wave period. From my point of view, it is important to know to which waves the analysis is sensitive, so which waves come out of the analysis with which amplitude. It would be great if the authors could include a similar figure (but for delta t = 10 min instead of 2 min since the difference between the images is 10 min) or at least the corresponding information (e.g., waves with periods between 20 min and 60 min are enhanced by a factor between 1 and 2, the shorter the period the stronger the enhancement) also in the manuscript. This makes it easier to find out which waves are addressed in this study and facilitates comparisons with future studies.

- We have provided a plot in the appendix A for the time resolution of our imager (lines 475-480 in the revised paper draft).

[Figure]

Figure 1 - Effect of time difference image filter on gravity wave amplitudes of various periods and acquisition times ($\delta t$). The black thick line corresponds to the acquisition time of our imager during SIMONe--2018.

- The plot is derived by applying the following equation

$$I'_{TD}/I' = 2 \sin(omega \cdot dt / 2) \qquad (1)$$

where $I'_{TD}$ is the TD wave amplitude, $I'$ is the real wave amplitude (prior TD filtering), omega is the wave frequency, and dt is the time resolution (10 minutes for our imager; dark continuous line in the Fig. 1 above).

• The calculation of TD images leads to a change in the amplitude of the waves (in the TD images compared to the original ones). The amplitude influences the FFT and therefore also the result of the cross-correlation analysis. The authors use the values of the cross-correlation analysis, e.g. for the derivation of the momentum flux. That means the calculation of TD images influences the momentum flux as far as I understand the algorithm. Has this been taken into account, for example, in the calculation of the error bars or is the effect negligible?

- The reviewer is correct. The TD filter affects the amplitude obtained from the cross-spectra. This issue is taken care of prior any wave parameter is computed. We estimate the correct wave amplitude I' by using equation (1) above.

Further comments:
p.15, l. 443 The authors write here that they restrict the data to 174 x 174 km², then, they calculate the TD pictures and apply a 2D FFT with a Hanning window on them (which makes the FoV smaller).

- The Hanning window does not make the FOV smaller, meaning the output frame size is the same as the input frame size. The window treats the edge effects due to the limited FOV size. We apply a symmetric 2D Hanning window (same size of the image) to the image frame prior calculating the FFT. The Hanning window tapers down from 1 at the center of the image to ~0 at the edges following the equation

$$w(n)=0.5(1-\cos(2\pi n/N)); \quad 0{\leq}n{\leq}N; \quad \text{window length = N+1}$$

In section 3.1, they define short-scale waves as waves with wavelengths shorter than 725 km since this is the diagonal of their original FoV (512 x 512 km²). I am aware of the fact that the FFT can also deliver wavelengths which are larger the side length or the diagonal of the reduced FoV (so 174 x 174 km²)

- See our comment above please.
- We agree the FFT can deliver wavelengths that are larger than the diagonal length. Please read our reasoning below (reply to your second question).

, however, if the waves do not fit completely into the FoV, the amplitude one gets from the FFT is reduced (the larger the wavelength, the worse the effect). The application of a Hanning window does not improve the situation since it makes the FoV smaller.

- The Hanning window helps in avoiding fake spectral peaks and aliasing. But the reviewer is correct, if the wave does not cover entirely the FOV, it is not possible to recover the full wave amplitude, although we have chosen the analysis window FOV to minimize that issue.

First question: Did I get the authors right that they used the FFT applied on the 174 x 174 km² FoV to derive wavelengths of up to 725 km? In figure 5 they show a maximum horizontal wavelength of 150 km but in the manuscript they state that the short wavelengths reach 725 km at maximum.

- The nominal FOV is determined by the dewarping operation to remove the fisheye lens distortions and mapping the image into an uniform geographical coordinate system. By default, we map the image into 512x512 km^2 FOV into a 512x512 pixel^2 frame, giving a pixel resolution of 1km/pixel. In this frame, the diagonal measures sqrt(512) ~725 km and defines the limit of the largest horizontal wavelength (lh) seen. In that case, the wave would cover the entire FOV. In general, waves seen have much smaller horizontal scales, and the analysis window of 174x174 km^2 was chosen by taking into consideration this fact, meaning the analysis window is well calibrated to capture most of the waves. If one do the analysis manually using the entire FOV, the wave scales observed will be in the output range the auto-detection method that is, lh<sqrt(174) km.

Second question: does a reduction of amplitude depending on the horizontal wavelength influence the results and if yes and this effect is not negligible, have the authors taken that into account?
- As stated above, we have taken care of the TD filter effect on the amplitude. It is different of what the reviewer wants to know, which is the FFT effect on the amplitude for waves with lh larger than the FOV diagonal length. The answer for that is: partially. That means, the analysis window is small enough to permit waves covering the entire area, since the horizontal extension of the wave packet is ~174x174 km^2 or smaller. By horizontal extension we mean the area covered by the wave oscillatory structure.
- If the horizontal extension of the wave package is larger than that, but lh<sqrt(174) km than we can obtain the wave amplitude fairly well, but we do not track departures of that case as we hypothesize that does not occur frequently based on observations.

Minor points:
p. 1, l., 14 relative amplitudes: relative to what? : corrected.
p. 5, l., 135 It's probably meant a rectangular window not a Gaussian one or something like that. It would be great if this info could be added. : corrected.
p. 5, l., 145 there needs to be "day" after "each" : corrected.
p. 5, l. 127 in the first version of the manuscript: I asked the authors to concretize the lapse rate they mean and they asked me whether I would like them to explicitly give the value of the lapse rate. No, it would just be good if the authors could insert "atmosphere" before "lapse rate" (in the line before the authors mentioned the atmosphere lapse rate and the adiabatic lapse rate) to make it clearer for the reader. : understood. Thanks for the clarification.
Table 3 I appreciate the error bars, but could the authors please also them to all components? : corrected.

Some typos:
p.5, l., 137 the instead of The: corrected.
p. 6, l. 161 simulated instead of simulate: corrected.
p. 9, l. 262 Probably 88–98 km instead of 86–98 km: corrected.
p. 14, l. 416 second lambda_z must be a lambda_h: corrected.

p.15, l. 439 Replace "." with ",": corrected.
p.16, l. 477 c_v instead of cv: corrected.

---

## Author Response (AR3)

Editor Decision: Publish subject to minor revisions (review by editor) (18 Jun 2021) by William Ward

Comments to the Author:

I have carefully examined the author's response to the latest comments from the reviewers. For the analysis of the larger scale waves and minor comments, their response is fine but they have not adequately addressed some of the issues associated with the analysis of the shorter scale waves. Unfortunately, these relate to the basic analysis approach that the authors use to determine these small scale gravity wave parameters and hence also impact the interpretation of their observations. These issues are outlined below.

1) Reviewer 1 commented on the lack of waves with intrinsic periods less than 20 minutes. The authors are correct that by shifting the images by the background wind prior to the analysis puts their analysis into a zero background wind framework. However, because their sampling time is 10 minutes, there will be aliasing of waves with periods less than 20 minutes into their analysis which have not been accounted for. This is because the technique used doesn't distinguish the propagation direction correctly and waves with periods that are harmonics of 10 minutes (i.e 10 minutes, 5 minutes) will not be seen since they will be observed as stationary waves. The fact that the exposure time is 2 minutes is not relevant to this aliasing issue.

The authors acknowledge that there is an ambiguity in the wave direction which cannot be directly resolved with the phase of the cross correlation but claim (Appendix B) that this can be resolved by choosing delta theta < 0 as this represents time progressing forward (ll 460 - 461). However, the sign of this angle depends on the propagation direction so this ambiguity remains. Note that Tang et al., [2005] whose analysis technique the authors follow, (page 105, first paragraph after Figure 2), also acknowledge the presence of this ambiguity but are able to resolve it since the period between images is two minutes. This time is short enough that it is unlikely that the gravity waves are able to travel more than half a wavelength between images. With the 10 minute sampling time associated with the data that authors are discussing this argument is no longer true. This issue would explain the issue that this reviewer noted. Other authors have observed numerous gravity waves with periods less than 20 minutes (Taylor et al., JGR, 1997; Li et al., 2018) and it is unlikely that this would not be the case for the location where the authors are making their measurements. This issue is likely the reason for the discrepancy between the observations of Li et al., 2011 and the author's observations which the authors comment on (lines 277 to 305).

Resolution of this issue in the context of this paper is difficult as it is fundamental to the author's calculation of the momentum flux and characterization of the waves during the observation period. Possibly acknowledging the observational filter and stating that the results are restricted to a subset of gravity waves would be the best way to proceed.

The filtering effect pointed by the editor is factual and can be verified in Fig. 11. However, figure 11 also shows that waves of 40 min will have finite amplitude. Observe in Fig. 11 that the amplitude decays fast for waves with a period <20 min and becomes zero for a period equal to 10 min. Of course, having a higher time resolution between sequential airglow images is better, as one can verify on Li et al. (2011) for a time resolution of 2 minutes when they could detect waves in the Brunt-Vaisala period range and above. Ideally, a better airglow experiment would rely on having all-sky imagers dedicated to each emission. This difficulty manifests here as the autodetection method's inability to capture waves of periods shorter than the integration time. Therefore, we have followed the editor's suggestion and added this discussion to lines 485-490.

2) Reviewer 1 also comments on the author's claim that the wave only lasts a short time. In response the authors use values of the vertical phase velocity to argue that a wave will pass through an airglow layer in less than 30 minutes. This argument is incorrect. The vertical phase velocity is not the appropriate parameter to determine the residence time of a particular wave in the field of view of the imager.

Typically several bands or wavefronts are seen in airglow images (as is shown in Figure 3 of this paper) and the 2D Fourier analysis identifies these bands as a wave with particular kx and ky. The airglow image provides a horizontal cut across a wave packet that is travelling with a group velocity perpendicular to and with a velocity magnitude different from that of the phase velocity. Hence the corrugated feature identified as a wave will remain in the image until the wave packet travels through the airglow layer. The actual residence time of a wave in the image requires knowledge of the group velocity and the scale of the wave package. The authors have not used this information (and are probably not able to) to determine the residence time of the wave.

This impacts the interpretation of the authors results and requires that the paragraph starting on line 184 needs to be rewritten. They can state that their convention is to count each set of three images where a wave is identified as a wave but need to acknowledge that they are likely multiply counting a particular wave packet with this convention.

We have followed the editor suggestion and rewrote the paragraph. In lines 185-190 one can read now: "It is possible that the observed wave events represent waves independent from one another because the observed waves have relatively long vertical wavelength and propagate vertically fast under weak horizontal winds. However, we recognize that this is not always the case and, as the oscillations slow down as they propagate vertically, their residence time within a given airglow layer could be long. Therefore, some of the detected waves could have been counted twice while evaluating the average momentum flux and other wave statistics.".

3) Reviewer 2 questions the amplitude determination of the small scale waves. The authors have responded appropriately to these comments. Never-the-less it would improve the paper if a few sentences were devoted to this in the main body of the text.

We have followed the editor suggestion and added the following to lines 175-180: "The auto-detection method relies on three sequential airglow frames to obtain two time-difference images used in the cross-spectral analysis to obtain gravity wave parameters. The calculation of time-difference images leads to a change in the amplitude of the waves (in the TD images compared to the original ones). The amplitude influences the Fourier analysis and therefore also the result of the cross-correlation. However, this issue is properly taken care of by restoring the amplitude of the waves as seen in the original images. Further details about this correction and the auto-detection method is found in Appendix B.".